

# 1 Microphysical properties of various precipitation systems worldwide 2 classified via objective methods based on dual-frequency 3 precipitation radar observations

Yujia Zhang[1,2] Xiaodong Zhang[1,2], Xiang Ni[1,2]
[1]Chongqing Jinfo Mou ntain Karst Ecosystem National Observation and Research Station, School of Geographical Sciences,
Southwest University, Chongqing, China
[2]Chongqing Engineering Research Center for Remote Sensing Big Data Application, School of Geographical Sciences,
Southwest University, Chongqing, China
*Correspondence to*: Xiang Ni (nixiang@swu.edu.cn)





**Abstract.** Microphysical properties play crucial roles in physical processes related to the development of precipitation. In this study, Global Precipitation Measurement (GPM) dual-frequency precipitation radar (DPR) data were processed to demonstrate the microphysical properties of different precipitation systems (PSs) that are objectively classified with the k-means clustering algorithm. Four types of regular/non-extreme PS (high-latitude shallow PS, subtropical shallow PS, moderate PS, deep PS) and four types of extreme PS (extreme deep PS, strong PS, extreme strong PS, and marine extreme PS) were recognized. These eight types of PS exhibit differences in spatial-temporal features and convection characteristics, such as storm height, rain intensity, and vertical structures. For example, with the highest radar echo top and the largest mean mass-weighted mean diameter ($D_m$), the extreme strong PS mainly locate over tropical continent, while the high-latitude shallow PS have the least precipitation rate and mean normalized intercept parameter ($N_w$) values. The relationships between convection features and microphysical properties also vary among the eight types of PSs. For extreme PS, maximum precipitation rate near the surface generally exceeds 100 mm h$^{-1}$ and balanced breakup and coalescence processes play a dominant role compared with non-extreme PS. In contrary, the coalescence processes dominate near the surface in two types of shallow PS. These results highlight the diversity of global precipitation microphysics and emphasize the necessity of global studies to increase the understanding of precipitation processes.





## 1. Introduction

The microphysical characteristics of precipitation provide crucial information for describing precipitation. The deficiency of precipitation microphysical parameterization schemes is a significant factor contributing to precipitation errors in weather and climate models (Snook and Xue, 2008). Accurately obtaining spatial and temporal distributions and variations in precipitation microphysical parameters is essential for understanding the physical processes of precipitation, increasing the accuracy of quantitative precipitation estimation (QPE), and evaluating microphysical parameterizations in models (Chen et al., 2011; Zhang et al., 2023). Currently, observations and characteristics of precipitation microphysics at the global scale remain lacking because of the limited number of observation approaches.

The drop size distribution (DSD) is a typical metric for depicting precipitation microphysics. DSD features can be derived from observations obtained via disdrometers, ground-based radar instruments, and space-based radar instruments. In radar instruments, the interaction of electromagnetic waves with hydrometeors is used to retrieve DSD parameters (Marzuki et al., 2023), whereas disdrometers measure raindrop counts to directly obtain DSDs at the surface. Disdrometers provide only point measurements at specific levels and cannot measure the vertical structure of DSDs. Moreover, disdrometers have not been deployed globally, especially over the ocean. Although ground-based radar instruments can measure the three-dimensional structure of precipitation, they can only be used in limited areas, and their observation accuracy is significantly affected by the terrain conditions within the observation area (Dai et al., 2020). In contrast, space-based radar instruments can provide the vertical structures of DSD parameters worldwide. This study focused on the microphysical characteristics of various precipitation systems (PSs) worldwide. Compared with other instruments, space-based radar instruments are the most suitable for researching global precipitation microphysics.

In 1997, the Tropical Rainfall Measuring Mission (TRMM) satellite was launched by the National Aeronautics and Space Administration (NASA) and the Japan Aerospace Exploration Agency (JAXA). The precipitation radar (PR), which operates in the Ku-band (13.8 GHz), was carried by the TRMM (Iguchi et al., 2000). This marked the beginning of the observation of precipitation microphysics via space-based radar instruments. Notably, DSD parameters were retrieved from the radar reflectivity measured by the PR with the assumption that the DSD can be characterized by the diameter parameter itself (Iguchi et al., 2000). As a result, the DSDs obtained via retrieval exhibited large errors. In 2014, NASA and JAXA successfully launched the Global Precipitation Measurement (GPM) Core Observatory (GPM-CO). The GPM-CO carried the first spaceborne dual-frequency precipitation radar (DPR) system, operating in the Ku and Ka bands (13 and 35 GHz, respectively) (Skofronick-Jackson et al., 2017). The differential scattering during rainfall at these two frequencies is directly related to the size of raindrops (Gatlin et al., 2020). Via the use of this characteristic, $D_m$ and $N_w$ can be retrieved. The retrieved DSD parameters have been verified with ground-based observations and are better than those obtained via the TRMM PR algorithm (Sun et al., 2020). In addition, validation studies have confirmed the feasibility of using DPR observations for DSD parameter analysis (D'Adderio et al., 2018; Peinó et al., 2024). Peinó et al. (2024) used observational data from seven Parsivel disdrometers across different topographic zones in the western Mediterranean to validate GPM





DSD products. They reported that the GPM DPR products effectively captured the variations in DSDs observed under
different rainfall intensities. Therefore, GPM DSD products have been widely employed to investigate the microphysical
characteristics of precipitation in the literature (Wen et al., 2024, 2023) .
However, previous studies involving GPM DSD products have focused mainly on specific locations or weather systems. For
example, Li et al. (2024) studied the vertical structure and DSD characteristics of different precipitation types during the
rainy season over South China and reported that the precipitation type and intensity affect the DSD parameters. In their study,
under the same precipitation intensity, shallow convective precipitation exhibited the smallest $D_m$ and largest $N_w$ values,
whereas deep convective precipitation exhibited the opposite phenomenon. Additionally, regarding stratiform precipitation,
for PR > 3.5 mm h$^{-1}$, $D_m$ slightly increased, and in regard to shallow convective precipitation, $D_m$ remained at approximately
1.3 mm for PR > 2 mm h$^{-1}$. Similarly, Wen et al. (2023) analyzed the seasonal variations in the vertical structure of
precipitation microphysics in East China. They reported that the spatial distributions of $D_m$ and $N_w$ demonstrate obvious
seasonal variations and that there are more small raindrops in convective precipitation in autumn and winter than during the
other seasons. These studies revealed the variations in microphysical characteristics across different seasons and rainfall
types. Additionally, regarding weather conditions, regional variations in the precipitation characteristics of tropical cyclones
(TCs) have been investigated over the North Indian Ocean (Kumar et al., 2023). Research has revealed that the nature of
microphysical processes largely influences the growth of droplets in convective and stratiform rain. Wu et al. (2022)
investigated the DSD characteristics of record-breaking Typhoon In-Fa (2021). Their findings revealed significant internal
and regional differences in the microphysical characteristics of typhoon precipitation. When different precipitation types
during Typhoon In-Fa were compared, convective precipitation ($N_w$ values ranging from 3.80 to 3.96 m$^{-3}$ mm$^{-1}$) exhibited
higher raindrop concentrations than did stratiform precipitation ($N_w$ values ranging from 3.40 to 3.50 m$^{-3}$ mm$^{-1}$).
Additionally, convective precipitation during Typhoon In-Fa indicated a greater (lower) raindrop concentration than that
during Typhoon Taiwan (Hainan), while the raindrop diameter was smaller than those during both Typhoons Taiwan and
Hainan. These studies primarily focused on the microphysical process and structure of various weather conditions, which
provided insight into the formation process of precipitation. At present, there are few studies on the microphysical
characteristics of large-scale and global PSs. On the one hand, as mentioned above, the DSD is influenced by numerous
factors, such as precipitation type and season. There may be multiple precipitation types and DSDs in one area. On the other
hand, few DSD datasets covering the whole world are available. Dolan et al. (2018) used twelve disdrometer datasets across
three latitudinal zones—high-latitude, midlatitude, and low-latitude zones—to analyze DSD spatial variability. They
reported that the DSD varies with latitude. At low latitudes, moderate $D_m$ values (1.5–2 mm) and large log10($N_w$) values (> 4
m$^{-3}$ mm$^{-1}$) dominated. At midlatitudes, high $D_m$ values and small $N_w$ values dominated. At high latitudes, low $D_m$ and large
$N_w$ values prevailed. Although the dataset covered a wide range of precipitation regimes, it could not capture all rain regimes.
Moreover, a regional DSD dataset cannot represent the DSD within a given latitudinal band because of the limitations of
disdrometers. Hence, in this study, GPM DSD products were employed to investigate the microphysical characteristics of
PSs at global scales.





This study aimed to classify different PSs on the basis of DPR observations via machine learning and to analyze the
microphysical characteristics of different types if PSs. The results could address regional DSD variability and increase our
understanding of the microphysical processes of different types of PSs. This study is organized in four sections. Section 2
provides detailed descriptions of the GPM data and machine learning models applied in this study. The main results are
presented in Section 3, and finally, a summary is given in Section 4.
**2. Data and methods**
**2.1. Data**
GPM observations cover the range from 65° S to 65° N (Hou et al., 2014; Tapiador et al., 2012). The GPM DPR operates in
the Ka and Ku bands, with a spatial resolution of approximately $5 \times 5$ km$^2$. The scanning of DPR is cross-track and has three
scan patterns: normal scanning (NS), matching scanning (MS), and high sensitivity scanning (HS) (Das et al., 2022). Since
the scanning pattern of the Ka-band was changed in 2018 (Awaka et al., 2021), the GPM 2A DPR (version 7) products
considered the changes in the Ka-band scan pattern with a more accurate precipitation estimation algorithm. The product
formats in version 7 have been changed from the original three types to two types: FS and HS. The FS product exhibits a
new format and is defined as a full-scan dual-frequency product with a 125-m distance resolution. Compared with previous
algorithms, the FS mode makes it possible for the first time to process a full-scan band of approximately 245 km in dual-
band mode (Awaka et al., 2021). Therefore, the FS type was adopted in this study.
In this study, five years (2018–2022) of 2A DPR products (version 7) were employed. The parameters used in this machine
learning model include DSD parameters ($D_m$ and $N_w$), near-surface precipitation rate (mm h$^{-1}$), attenuation-corrected radar
reflectivity (dBZ), reflectivity near the surface ($Z_{surf}$), and typeprecip (stratiform and convective precipitation pixels are
distinguished by the typeprecip parameter), and airTemperature (this parameter can be used to distinguish between snow and
rain).
**2.2. Precipitation system (PS)**
This paper presents a method based on the connected domain principle for identifying PSs similar to those contained the
widely used TRMM/GPM Precipitation Feature dataset (Liu et al., 2008, 2020). Similar to the Precipitation Feature dataset
(Liu et al., 2008), neighboring precipitation pixels, with a minimum precipitation rate of 0.1 mm h$^{-1}$, are grouped into a PS.
Each PS is required to have a minimum of four precipitation pixels.
The DPR can observe the three-dimensional structure of precipitation and DPR products include radar reflectivity
parameters and retrieved DSD parameters from 0 to 22 km with a range resolution of 125 m, resulting in a total of 176 layers
of data. Consequently, for each PS type, DSD and radar reflectivity parameters such as the maximum and average values of
each layer were calculated. The average $D_m$ and $N_w$ profiles were used for each PS, and if the profiles of the maximum $D_m$
and $N_w$ values in each layer were involved, MAX-$D_m$ and MAX-$N_w$, respectively, were used. Given the potential



relationships of the convective intensity with microphysical parameters, $Z_e$ in the product was employed to calculate the
maximum 20/30/40 dBZ echo top height (MAXHT20/30/40) for each type of PS (Liu, 2011; Liu et al., 2020; Ni et al., 2019;
Roy et al., 2020), the echo top height of the PS ($H_{top}$) (Arulraj and Barros, 2021), and other convective parameters. To
characterize the conditions of the PS, several additional features were calculated, such as the maximum precipitation rate
near the surface (the maximum precipitation rate of the precipitation pixels included in the PS) and the precipitation area (the
number of precipitation pixels contained in the PS). Considering that the GPM satellite exhibits a higher observation
frequency in high-latitude regions (approximately 2–3 times that at the equator), the original dataset is prone to oversampling
in these areas, which can introduce bias. To construct a balanced dataset suitable for clustering analysis, this study
implemented a homogenization for the sampling. Specifically, the satellite's observation frequency was calculated as a
function of latitude, and sample size for each latitude was adjusted using the ratio of its frequency to that at the equator.
Subsequently, precipitation systems were randomly selected from each latitude to ensure a consistent scaled sample size,
thereby effectively addressing the issue of uneven sampling. Finally, a total of 8,924,307 PSs were obtained for subsequent
analysis.

### 2.3. Methods


In this study, two distinct machine learning models, namely k-means clustering and principal component analysis (PCA)
were used. Both models were trained and evaluated via the Python scikit-learn package. These models are briefly described
below. The k-means algorithm is one of the most popular clustering algorithms among machine learning algorithms. It is one
of the most popular unsupervised clustering algorithms due to its efficiency (Jain, 2010). The algorithm follows a three-step
process. Initially, it aims to select initial cluster centers by randomly obtaining sample coordinates from the dataset and
assigning each sample to its nearest cluster center. Next, it computes the mean of all sample points assigned to each previous
cluster center to establish new cluster centers. Finally, the algorithm aims to evaluate the differences between the new and
old cluster centers. If differences are present, the last two steps are repeated until the cluster centers stabilize and no longer
shift (Jain, 2010).
PCA is a classical dimensionality reduction tool in machine learning (Gang and Bajwa, 2022). PCA is based on the linear
combination of target features to construct the principal subspace, and the variance is then employed to measure the
information content with the aim of identifying the linear subspace with the maximum variance (Marukatat, 2023). In
summary, PCA aims to transform numerous pertinent features into a comparatively limited number of irrelevant ones,
thereby retaining as much of the informational content of the original data as possible (Gang and Bajwa, 2022). Considering
that there are 176 vertical layers of GPM DPR products, if all DSD data were used as input parameters, the clustering effect
could be poor because of the high dimensionality. In this study, PCA was adopted to reduce the dimensionality of the data
while striking a balance between information loss and the optimal number of parameters to be retained (Festa et al., 2023;
Jolliffe and Cadima, 2016).



In this study, the maximum precipitation rate near the surface, $H_{top}$, the precipitation area, the proportion of stratiform
precipitation, the proportion of convective precipitation, the DSD parameters ($D_m$ and $N_w$) and the maximum radar
reflectivity parameter ($Z_e$) after dimensionality reduction via PCA were used as input parameters for the k-means clustering
algorithm. These parameters were selected based on their critical role in comprehensively characterizing the features,
structure, and microphysical processes of precipitation systems. Among them, the maximum surface precipitation rate and $Z_e$
reflect the intensity of the precipitation process and its echo characteristics, while the precipitation area directly characterizes
the spatial differences in both the vertical and horizontal distributions of the system. The $H_{top}$ not only reveals the vertical
distribution but also captures the top-level information of the precipitation cloud through the maximum reflectivity height.
Introducing the proportions of stratiform and convective precipitation facilitates the differentiation of precipitation types
generated by distinct mechanisms, thereby elucidating their evolution patterns and dynamic characteristics. Furthermore, the
DSD parameters ($D_m$ and $N_w$) effectively describe the size distribution of precipitation particles and their intrinsic physical
processes, providing an essential basis for an in-depth understanding of precipitation microphysics. Collectively,
constructing a multidimensional precipitation feature space with these parameters enhances the accuracy and robustness of
the clustering analysis.
The quality of clustering was evaluated by analyzing different clustering structures derived from the same dataset. The most
commonly employed performance metrics, such as the sum of squared errors (SSE), Davis Bouldin (DB) index, Calinski-
Harabasz (CH) Score (El Khattabi et al., 2024) and silhouette index, can be utilized to assess the effectiveness and quality of
clustering algorithms (Ay et al., 2023). In this case, the DB index was calculated by computing the average sum of the
intraclass distances between any two clusters divided by the distance between the centers of those two clusters and obtaining
the maximum value. The DB index can manage clusters of different sizes and densities with a high degree of robustness to
noise and outliers.
The DB index is calculated by computing the average sum of intraclass distances between clusters, divided by the distance
between their respective centers, with the final value determined by the maximum across all clusters. A lower DB index
indicates better clustering performance (Sowan et al., 2023). Additionally, the CH score, which assesses clustering
compactness and separation, was also considered. Higher CH scores indicates better-defined clusters. Algorithms with
clustering numbers ranging from 3 to 20 were executed, and the resulting change in the DB index and CH score was plotted
(refer to Fig. S1 in the Supplementary Material). The results show that when K = 8, the DB index reaches its lowest value,
while the CH score remains relatively high, indicating a well-balanced clustering structure. Therefore, the optimal number of
clusters is eight. Combining all the features of the PSs described in Section 3, the Cluster 1-8 could be regarded as four non-
extreme PS (high-latitude shallow PS, subtropical shallow PS, moderate PS, deep PS) and four extreme PS (extreme deep PS,
strong PS, extreme strong PS, and marine extreme PS), which are listed here for the convenience of understanding the
following context.





**3. Results and discussion**
**3.1. Global distributions**
Table 1 shows the statistics of various parameters for the eight types of PS. There numbers include abundant information and
verify the rationality of the objectively clustering algorithm. First, the numbers of the various types of PSs differed
significantly. The two types of shallow PSs (high-latitude shallow PS and subtropical shallow PS) accounted for 81.44% of
the total PS count. The proportions of deep and moderate PSs were 2.41% and 15.50%, respectively. The other four types of
PS are regarded as extreme PS (extreme deep PS, strong PS, extreme strong PS, and marine extreme PS) because their ratios
of the total PS are less than 1%, accounted for only 0.39%, 0.22%, 0.02%, and 0.01%, respectively. In the non-extreme PS,
MAXHT20 is generally positively related to the precipitation rate (Table 1). However, in the extreme PS, the correlation
between the extreme precipitation rate and MAXHT20 is not clear. For example, that the mean value of the maximum
precipitation rate in marine extreme PS was the highest among the eight types of PSs, although its MAXHT20 was less than
that in extreme strong PS and close to that in extreme deep PS. This result is consistent with other studies noting a weak link
between the heaviest rainfall and the highest storm top (Hamada et al., 2015). Although the convective intensity of extreme
deep PS is not significantly higher than that of deep PS, it exhibits a substantially larger precipitation area and maximum
precipitation rate.
High-latitude shallow PS was most prevalent at midlatitudes and high latitudes, where snowfall and sleeting are more
frequent than at low latitudes. Notably, high-latitude shallow PS were dominated by stratiform precipitation, with stratiform
pixels accounting for 88.63%. Meanwhile, approximately 86.60% of the PS exhibited surface temperatures higher than 0 °C.
A study confirmed that at high latitudes and in polar regions, more than 25% of precipitation falls as snow (Lerber et al.,
2018). This is consistent with the observations from high-latitude shallow PS. Additionally, an analysis of high-latitude
shallow PS by latitude revealed that with increasing latitude, the number of samples generally increased. Moreover, the
number of PSs with echo top heights less than 2.5 km increased with latitude. During the winter season at 65°S, PSs with
echo top heights below 2.5 km accounted for approximately 50% of the total PSs there. This is likely due to the influence of
the low surface temperature and weak convection (refer to Fig. S2 in the Supplementary Material).
Subtropical shallow PS primarily occurred over the ocean where is dominated by the subtropical high, with a relatively
limited degree of overlap with moderate PS and deep PS (Fig. 1). The mean MAXHT20 value in subtropical shallow PS was
only 3.29 km, and the proportion of convective precipitation was the highest among all the types of PSs, exceeding 90%.
Compared with those of the other PSs, subtropical shallow PS exhibited the smallest precipitation area. Moreover, it was
rarely found over land. These results support the conclusion that subtropical shallow PS is associated with isolated shallow
convection over the ocean, which has been the topic of interest in previous studies (Chen and Liu, 2016; Chudler et al., 2022;
Houze Jr. et al., 2015).
The geographic distribution patterns of deep PS and moderate PS were approximately the same (Fig. 1). The number of
occurrences in the maritime continent (MC), Indian Ocean, Atlantic Ocean, Amazon rainforests and Pacific Ocean were



relatively high. There regions are generally influenced by the Intertropical Convergence Zone (ITCZ). Nevertheless, the deep
PS has higher land percentage. The mean values of the maximum precipitation rates in moderate PS and deep PS were 6.21,
35.94 mm h$^{-1}$, respectively, whereas those of MAXHT20 were 7.03 and 11.89 km, respectively.
Strong PS, extreme deep PS, extreme strong PS, and marine extreme PS demonstrated low sample sizes. However, their
precipitation areas are significantly larger than non-extreme PS (Table 1). The location of extreme deep PS is similar with
moderate and deep PS, with larger values for most parameters. In the extreme strong PS, the proportion of land pixels
reaches 81%, with significant concentrations in near-equatorial Africa, America, India, the southeastern U.S., and South
America. The average maximum precipitation rate in extreme strong PS was 156.37 mm h$^{-1}$, and MAXHT40 reached 12.32
km, which is the highest among all the types of PSs. The high MAXHT40 value indicates strong updraft in the middle
troposphere, which is favorable for hailstone formation. Therefore, the spatial distributions of hailstorms in extreme strong
PS were very similar to those of hailstorms with large hailstones on the ground (Marra et al., 2017). Marine extreme PS was
primarily situated in the near-equatorial marine region, with only 943 PSs and 90% is over the ocean. The mean maximum
precipitation rate in marine extreme PS was 178.30 mm h$^{-1}$, ranking first among the eight types of PSs. Although the
MAXHT20 value in marine extreme PS reached 12.81 km, the MAXHT40 value in marine extreme PS was approximately
half of that in extreme strong PS, indicating low convection activity in the middle and upper levels. Oceanic extreme PS
(extreme deep PS and marine extreme PS) with a high fraction of ocean pixels, exhibit a significantly larger precipitation
coverage area than continental extreme PS (strong PS and extreme strong PS). This spatial distribution aligns with previous
findings that the most extensive PS are predominantly located in oceanic regions. Furthermore, continental extreme PS
display markedly stronger convective intensity. This disparity is largely attributable to the observation that the heaviest PS
generally occur over tropical land, the Western Pacific warm pool, the North American Great Plains, and Argentina, whereas
the most severe convective storms are predominantly observed over continental areas (Liu and Zipser, 2015).




**Figure 1.** Spatial distributions (2° × 2°) of the PS counts from 2018 to 2022




**Table 1.** Precipitation parameters for the different types of PSs. (* indicate that in high-latitude shallow PS and subtropical shallow PS, approximately 80% of the samples do not reach 40 dBZ. Therefore, the mean MAXHT40 for these samples is recorded as 0.)

| | high-latitude shallow PS | subtropical shallow PS | Moderate PS | deep PS | extreme deep PS | strong PS | extreme strong PS | marine extreme PS |
|---|---|---|---|---|---|---|---|---|
| Mean MAXHT20 (km) | 3.40 | 3.29 | 7.03 | 11.89 | 12.67 | 15.39 | 17.21 | 12.85 |
| Mean MAXHT30 (km) | 2.63 | 2.67 | 5.11 | 8.65 | 8.52 | 13.68 | 16.31 | 9.18 |
| Mean MAXHT40 (km) | 0.00* | 0.00* | 3.44 | 5.53 | 5.71 | 8.64 | 12.32 | 6.04 |
| Stratiform percentages (%) | 88.63 | 9.46 | 54.38 | 53.22 | 69.90 | 57.42 | 53.02 | 66.83 |
| Convective percentages (%) | 5.85 | 89.95 | 42.83 | 44.52 | 28.16 | 39.91 | 44.06 | 31.56 |
| Land percentages (%) | 21.61 | 6.97 | 27.96 | 42.31 | 15.61 | 65.37 | 80.98 | 10.45 |
| Ocean percentages (%) | 78.39 | 93.03 | 72.04 | 57.69 | 84.39 | 34.63 | 19.02 | 89.55 |
| Mean precipitation (mm h$^{-1}$) | 1.60 | 2.35 | 6.21 | 35.94 | 156.67 | 135.46 | 156.37 | 178.30 |
| precipitation Standard deviation (mm h$^{-1}$) | 1.63 | 1.92 | 8.89 | 50.44 | 98.44 | 106.95 | 103.50 | 98.61 |
| Number of samples | 4,184,547 | 3,083,077 | 1,383,261 | 215,611 | 34,982 | 19,790 | 2,096 | 943 |
| Mean precipitation area (km$^2$) | 610.57 | 239.23 | 2761.46 | 7009.37 | 37076.93 | 18485.91 | 22521.51 | 36044.11 |
| >273.15 K frequency (%) | 86.60 | 99.16 | 99.83 | 99.97 | 99.97 | 99.99 | 99.99 | 100.00 |
| 2.5 km Mean MAX-log10($N_w$) [m$^{-3}$ mm$^{-1}$] | 3.47 | 3.70 | 4.06 | 4.49 | 5.20 | 4.72 | 4.88 | 6.07 |
| 2.5 km Mean MAX-$D_m$ [mm] | 1.03 | 1.17 | 2.26 | 2.82 | 2.71 | 3.04 | 3.11 | 2.61 |
| 2.5 km Mean log10($N_w$) [m$^{-3}$ mm$^{-1}$] | 3.23 | 3.45 | 3.36 | 3.39 | 3.83 | 3.36 | 3.35 | 4.45 |
| 2.5 km Mean $D_m$ [mm] | 0.85 | 0.89 | 1.36 | 1.50 | 1.30 | 1.61 | 1.71 | 1.32 |

**3.2. Global distributions of microphysical features**

Fig. 2 and Fig. 3 show the global distributions of the microphysical parameters for the eight types of PSs. To avoid the influence of ground clutter, in each PS, the mean $D_m$ and $N_w$ values at 2.5 km above the ground surface were analyzed. Notably, there was a significant degree of spatial heterogeneity in each panel. The general conclusion is that continental PSs exhibit a higher $D_m$ than do oceanic PSs. Usually, continental rainfall is associated with high convective activity in which



clouds produce large raindrops. Over land, small raindrops are lifted by updrafts, whereas large raindrops are formed from
the melting of larger ice crystals. In contrast, oceanic rainfall is accompanied by the formation of weak updrafts and the
development of a low melting layer, which impedes the formation of large raindrops and results in a high concentration of
small raindrops (Saha et al., 2022; Seela et al., 2018). Moreover, $D_m$ decreases with increasing latitude, a trend that is
especially notable in high-latitude marine regions (refer to Fig. S2c in the Supplementary Material). Cha et al. (Cha and Yum,
2021) noted that snow primarily comprises small particles (diameter < 1 mm). In high-latitude shallow PS, snowfall may
become more frequent from the middle to high latitudes, which can result in a decrease in $D_m$. Notably, the height and
thickness of the melting layer may influence raindrop growth (Hu et al., 2024). With increasing latitude, the melting layer
becomes thinner, thus reducing the conditions necessary for raindrop growth, which may lead to the formation of a larger
number of small raindrops. In the oceanic regions within subtropical shallow PS, the higher sea surface temperature in the
tropics is more conducive to convection formation and development. Moreover, $D_m$ varies among the eight clusters in a
specific region. For example, in the Amazon region, moderate PS exhibits a lower $D_m$ than deep PS does.
Similar to $D_m$, there is a distinct contrast in $N_w$ between continents and oceans. Continental rainfall is usually associated with
the cold rain mechanism, whereby raindrops grow as ice particles. In contrast, oceanic rainfall is associated with a warm rain
regime, in which raindrops grow via a collision-agglomeration mechanism. Consequently, $N_w$ over land is less than that over
oceans (Suh et al., 2016). For the same PS, $N_w$ is high in areas with small $D_m$ values and conversely low in areas with large
$D_m$ values. For example, in extreme deep PS, the $D_m$ value over the eastern near-equatorial Pacific Ocean, which reaches
approximately 1.18 mm, is smaller than that of the other oceanic regions. However, $N_w$ is significantly greater than those in
the other regions. In strong PS, the $D_m$ values in near-equatorial Africa and the eastern United States are greater than those in
other regions, but the $N_w$ values are lower than those in other regions. It is possible that $D_m$ and $N_w$ may be negatively
correlated for the same PS.



**Figure 2.** Spatial distributions of the mass-weighted mean diameter ($D_m$) for the eight PS clusters at a height of 2.5 km.



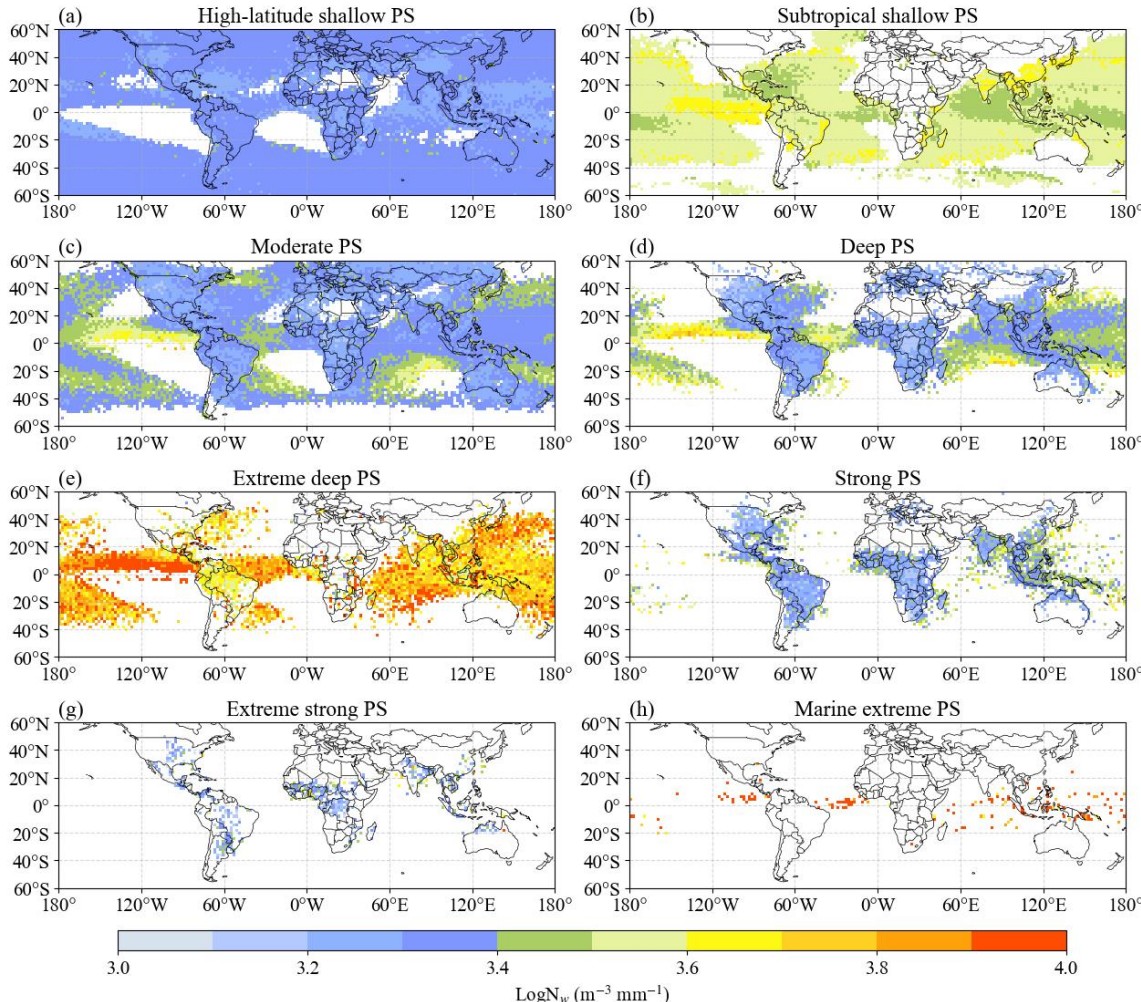


**Figure 3.** Similar to Fig. 2. but for $\log_{10}(N_w)$.

## 3.3. Vertical structure of the different PS types

The contoured frequency by altitude diagrams (CFADs) of $D_m$, $N_w$, and the maximum radar reflectivity for the eight clusters are shown in Fig. 4/5/6. Figure 4 shows the CFAD of the maximum radar reflectivity profiles. The results revealed high echo tops for deep PS, extreme deep PS, strong PS, and extreme strong PS, and low echo tops for high-latitude shallow PS and subtropical shallow PS. Extreme strong PS attained an echo top height greater than 18 km, and it also exhibited the strongest convection at the middle level. Its geographic distribution was exclusively terrestrial, which is consistent with other studies concluding that deep convective cores occur mostly over land (Houze Jr. et al., 2015). Extreme deep PS and marine extreme PS exhibited sharper decreasing trends from 6–12 km than that in extreme strong PS. Therefore, extreme strong PS encompassed a greater amount of supercooled liquid droplets or large ice–water vapor condensates produced by strong





convective updrafts than that in extreme deep PS and marine extreme PS (Jiang, 2012). Owing to the lack of strong updrafts
in extreme deep PS and marine extreme PS, the reflectivity rapidly decreased with height above the freezing level. Table 1
indicates that the land proportion of extreme strong PS was much greater than that of extreme deep PS and marine extreme
PS. Additionally, land indicates a dry adiabatic lapse rate, which results in greater buoyancy and allows for stronger updrafts
to lift ice crystals higher into the atmosphere. As a result, the maximum radar reflectivity in the middle levels at high
altitudes decreased more slowly in extreme strong PS. High-latitude shallow PS and subtropical shallow PS yielded low echo
tops of less than 6 km, indicating low convective intensity. Therefore, subtropical shallow PS could be identified as being
associated with isolated shallow convection over the ocean, especially the region dominated by the subtropical high.

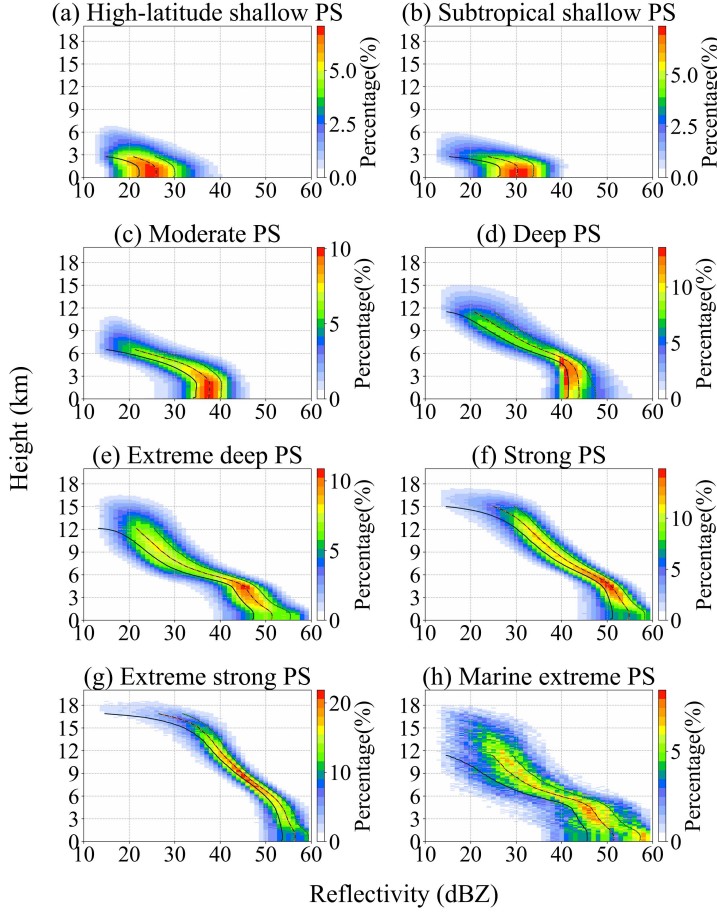




**Figure 4.** Contoured frequency by altitude diagrams (CFADs) of the maximum radar reflectivity for the eight distinct PS clusters. The solid lines indicate the 25th percentiles; the dashed-dotted lines indicate the 50th percentiles; the dotted lines indicate the 75th percentiles.

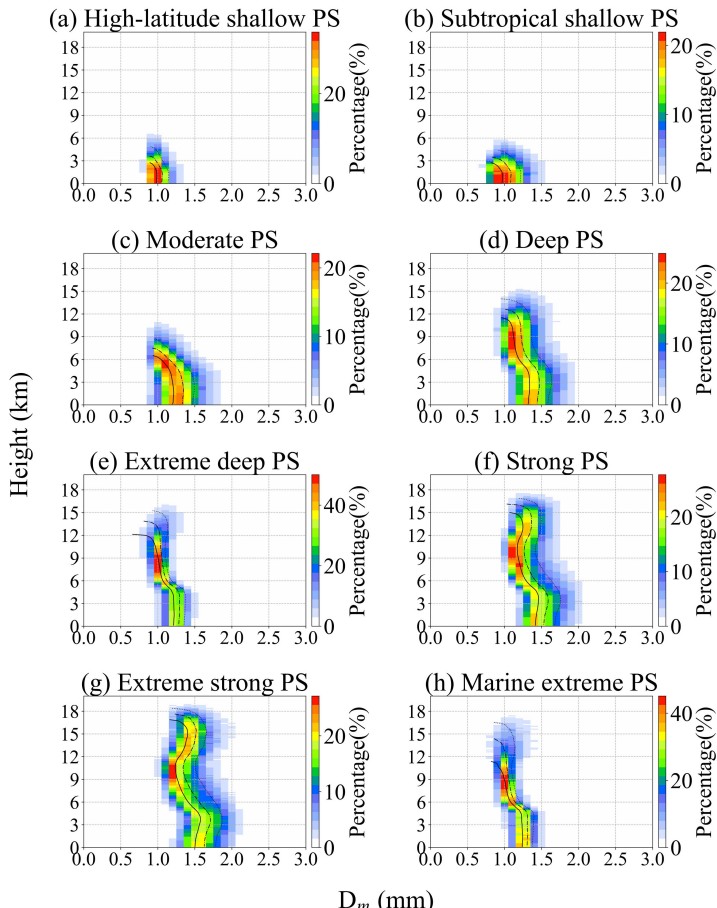

**Figure. 5.** Similar to Fig. 4, but for D$_m$.



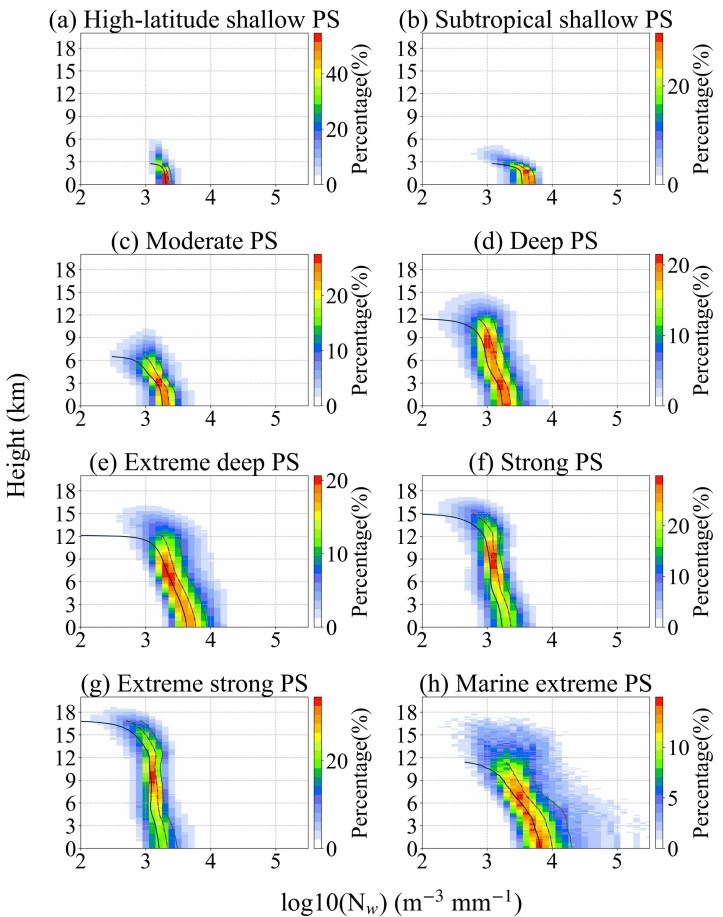

**Figure. 6.** Similar to Fig. 4, but for $\log_{10}(N_w)$.

Figure 5 shows the CFAD of $D_m$ for the eight types of PSs. Generally, deep convections (deep PS, extreme deep PS, strong PS, extreme strong PS, and marine extreme PS) produce different $D_m$ values in the regions above and below approximately 5 km. Moreover, strong PS and extreme strong PS exhibited wider distributions than those of extreme deep PS and marine extreme PS. For deep PS, strong PS, and extreme strong PS, $D_m$ below 4.8 km did not change much or slightly increased along with height, but the value decreased between 4.8 and 6.9 km. In extreme strong PS, the vertical structure of $D_m$ was more complex. Extreme strong PS exhibited three regimes according to the variations in $D_m$. The first regime was observed between 0 and 4.1 km, where $D_m$ increases with altitude. This is consistent with other papers involving the use of ground-based radar observations and reporting that $D_m$ of deep convective precipitation decreases with decreasing height near the surface (Marzuki et al., 2023). The observed decrease in $D_m$ may be related to the continued breakdown of large isolated





raindrops in the atmosphere. The second regime was observed above the freezing level, from 4.1 to 10 km, where $D_m$
decreases with altitude. In this regime, the updraft in deep convection was decreased (Uma and Rao, 2009). The decline in
updraft decreased the size of the particles that can be retained in the cloud. Finally, the third regime was observed between
10 and 18 km, where $D_m$ increases with altitude and where strengthened updrafts are often observed (Becker and
Hohenegger, 2021). Although both high-latitude shallow PS and subtropical shallow PS were shallow PSs, subtropical
shallow PS had a wider distribution of $D_m$ than high-latitude shallow PS. One possible reason is that in shallow oceanic
convection, the breaking of large raindrops broadens the DSD.
Figure 6 shows the CFAD of $\log_{10}(N_w)$ for the different types of PSs. In general, $N_w$ decreases with increasing altitude. The
distribution range of $N_w$ for shallow PSs was relatively small. Moreover, the $N_w$ distribution range of subtropical shallow PS
was larger than that of high-latitude shallow PS. Among PSs with intense convection, PSs with a greater proportion of land
coverage exhibited more concentrated $N_w$ values, whereas PSs with a greater proportion of ocean coverage exhibited higher
$N_w$ values. For example, the $N_w$ values of strong PS and extreme strong PS were smaller and narrower than those of ocean-
dominated deep PS, extreme deep PS and marine extreme PS. This finding is consistent with the conclusions of other studies
(Kumar et al., 2024). One possible explanation is that the slower updrafts over ocean regions result in higher concentrations
of smaller condensates at lower altitudes.
**3.4. DSD characteristics at a height of 2.5 km**
Figure 7a-h show the frequency distributions of the mean $D_m$ and $\log_{10}(N_w)$ values observed at 2.5 km above ground level.
The mean $D_m$ values for the eight types of PSs were 0.85, 0.89, 1.36, 1.50, 1.30, 1.61, 1.71, and 1.32 mm, and the
corresponding $\log_{10}(N_w)$ values were 3.23, 3.45, 3.36, 3.39, 3.83, 3.36, 3.35, and 4.45 $m^{-3}\,mm^{-1}$, respectively, as detailed in
Table 1. Generally, all the distributions shown in Fig. 7a-h greatly deviate from the parameters of continental convection and
maritime convection defined by Bringi et al. (2003). One reason is that the mean values of $D_m$ and $N_w$ for one PS were
considered here, whereas Bringi et al. (2003) separated the observation samples into stratiform and convection samples.
Moreover, the DSDs observed by disdrometers are generally cumulative observations of a single storm at one fixed location
and differ from the results for each PS in this study, which represent the instantaneous occurrence of a storm. With the most
intense convection at the middle level, extreme strong PS was the closest to continental convection (Fig. 7d), whereas marine
extreme PS was the closest to maritime convection (Fig. 7e). For most PSs, $D_m$ and $N_w$ were negatively correlated, with
greater dispersion of $D_m$ than that of $N_w$. Moreover, the shallow PSs, such as high-latitude shallow PS, exhibited lower $D_m$
and $N_w$ values and more concentrated distributions than those of the deep PSs, such as those in deep PS.





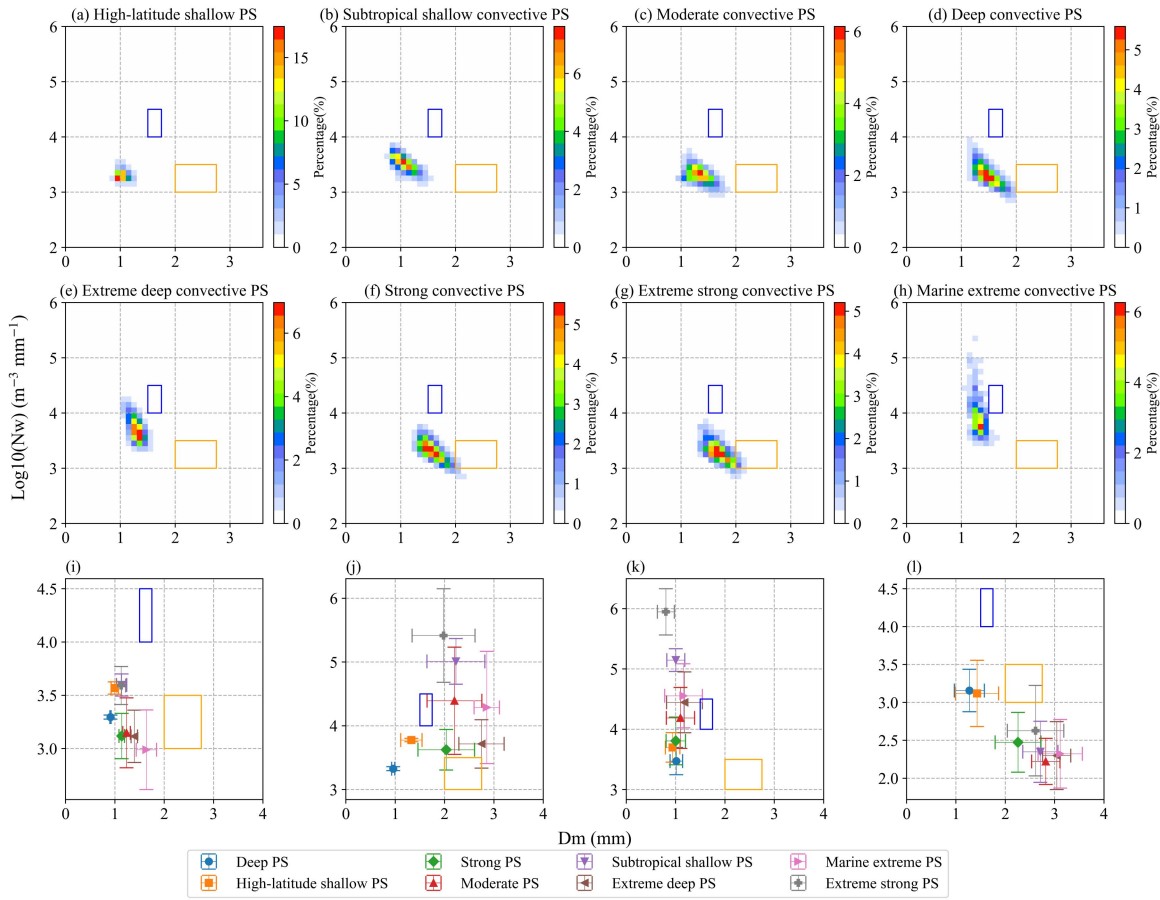

**Figure 7.** (a-h) Two-dimensional frequency distributions of $D_m$ and log10($N_w$) at a height of 2.5 km, and (i-l) statistical values of log10($N_w$) and $D_m$ for each PS (the bar indicates one standard deviation). (i) Mean values of $D_m$ and log10($N_w$), (j) MAX-$D_m$ and MAX-log10($N_w$), (k) MAX-log10($N_w$) and $D_m$ at its corresponding position, and (l) MAX-$D_m$ and log10($N_w$) at its corresponding position for each PS. (the blue and orange rectangles denote the maritime and continental convective clusters, respectively, in $D_m$ and log10($N_w$) space from Bringi et al. (2003)).

To further compare the mean $D_m$ and $N_w$ values of the different clusters, Figure 7i shows a summary of the mean $D_m$ and $N_w$ values, with the standard deviation for each type of PS. Marine extreme PS showed a significant abnormal value of $N_w$, whereas the $N_w$ value of extreme deep PS slightly deviated from those of the other PS. However, if only three extremely deep PSs with the highest echo tops, as detailed in Table 1 (strong PS, extreme strong PS, and marine extreme PS), were considered, it could be concluded that the larger the $D_m$ value is, the smaller the $N_w$ value. Moreover, the other PSs exhibited very similar $N_w$ values. These results might suggest that in deep convection, the DSD parameters at the near-surface level are related to convection intensity parameters. Ni et al. (2019) revealed that the dual-frequency ratio between the Ku and Ka



bands at 12 km was positively correlated with intensity parameters such as MAXHT20/30, partly because stronger updrafts
could hold larger ice particles in clouds. However, in swallow convection systems such as those in high-latitude shallow PS
and subtropical shallow PS, the relationship did not hold, which rendered the relationship between microphysical parameters
and convection parameters complex.
Note that although the mean $D_m$ and $N_w$ values represent the overall features of DSDs in one PS, they do not capture the
variety of DSDs in each PS. For example, the DSD might differ between convective and stratiform regions, where the $N_w$–
$D_m$ relationships might vary. To comprehensively demonstrate the microphysical features of PSs, Figure 7j shows the mean
MAX-$D_m$ and MAX-$N_w$ values of each PS at 2.5 km above ground level. For extreme PS (extreme deep PS, strong PS,
extreme strong PS, and marine extreme PS), a negative correlation was found between MAX-$D_m$ and MAX-$N_w$, similar to
the mean $D_m$ and $N_w$ values shown in Fig. 7h. However, for the non-extreme PS, MAX-$D_m$ and MAX-$N_w$ exhibited positive
correlations. A similar relationship is also shown in Fig. 7k, which suggests a relationship between MAX-$N_w$ and the
corresponding $D_m$ value in the MAX-$N_w$ pixels of each PS. Nevertheless, as shown in Fig. 7k, the $D_m$ values of all eight
types of PSs were very close. Nevertheless, it could be also found that in the non-extreme PS the $D_m$ increases with MAX-$N_w$,
while in the extreme PS, the $D_m$ decreases with MAX-$N_w$. Figure 7l shows the relationship between MAX-$D_m$ and the
corresponding $N_w$ value in the MAX-$D_m$ pixels of each PS. Interestingly, for all eight types of PSs, MAX-$D_m$ and $N_w$ showed
significantly negative correlations. Note that MAX-$D_m$ and MAX-$N_w$ in Fig. 7j are the maximum values for one PS and
usually do not occur in the same pixel. Figure 7k-l show the $N_w$–$D_m$ relationship observed at the same location. Overall, the
conclusions generally indicated that deep PSs yield larger MAX-$N_w$ or MAX-$D_m$ values than shallow convection PSs do.
Overall, extreme PS exhibited negative correlations between $N_w$ and $D_m$, whereas non-extreme PS demonstrated positive
correlations.
Ryu et al. (2021) analyzed DSDs during three types of heavy rainfall events with different rain intensities. They also
reported that $D_m$ increases with increasing rainfall intensity, whereas $N_w$ decreases with increasing rainfall intensity. In this
study, we saw a positive relationship between the increase in $D_m$ and MAXHT20 in extreme PS. However, extreme strong
PS attained the highest MAXHT20 value, but its precipitation rate was lower than that of extreme deep PS and marine
extreme PS. These results suggest a complex relationship between the microphysical parameters and convection features,
especially in deep and intense convection systems. Notably, in extreme convection, with strong convection at the top of the
storm, attenuation becomes notable at low storm levels, which might influence the retrieval of microphysical parameters. To
assess the impact of attenuation on the $D_m$-$N_w$ relationship, ground-based observations of microphysical properties from
disdrometers are needed. Finally, we considered the PS as a whole and did not account for the variations in the $D_m$ and $N_w$
values of each PS. The microphysical characteristics varied among different pixels. The mean or maximum values of $D_m$ and
$N_w$ only reflect part of the total process. Therefore, analyses on the basis of pixel-level observations would improve this
work.
To gain further insight into the primary microphysical processes associated with the various PS, we employed an
investigative approach analogous to that utilized by Kumjian and Prat (2014). To prevent the influence of ground-based





clutter, $\Delta Z_e$ and $\Delta D_m$ values were calculated as the difference between $Z_e$ and $D_m$ at 2 and 3 km above the ground.
Specifically, $\Delta Z_e = Z_e^{2km} - Z_e^{3km}$ and $\Delta D_m = D_m^{2km} - D_m^{3km}$ are calculated. Fig. 8 shows the frequency pattern of $\Delta Z_e$ versus
$\Delta D_m$ for the eight types of PSs. An increase (decrease) in $Z_e$ and $D_m$ indicates that coalescence (breakup) processes dominate.
Balanced breakup and coalescence processes result in an increase in $Z_e$ but a decrease in $D_m$. In contrast, a decrease in $Z_e$ and
an increase in $D_m$ are due to predominate evaporation or size sorting processes (Wen et al., 2023).
The microphysical processes of the different types of PSs were significantly distinct. Notably, the microphysical processes
were dominated by coalescence in the two types of shallow PS (Fig. 8a-b). Previous studies have demonstrated that high-
latitude shallow PS are more likely to experience the condensation of rain droplets into snow due to the low temperatures in
these regions. (Thompson et al., 2015). Meanwhile, the coalescence process plays an important role in tropical oceanic
shallow convective precipitation (subtropical shallow PS) as demonstrated by Li et al. (2024). Balanced breakup and
coalescence processes in the microphysical processes of extreme PS accounted for more than 40% of the total microphysical
processes, significantly exceeding other three types of microphysical processes. The microphysical processes may reach an
equilibrium state under high rainfall rates, in which the coalescence and breakup of raindrops are nearly balanced. Extreme
deep PS and marine extreme PS encompassed a higher percentage of coalescence processes than strong PS and extreme
strong PS did, whereas strong PS and extreme strong PS encompassed a higher percentage of breakup processes.

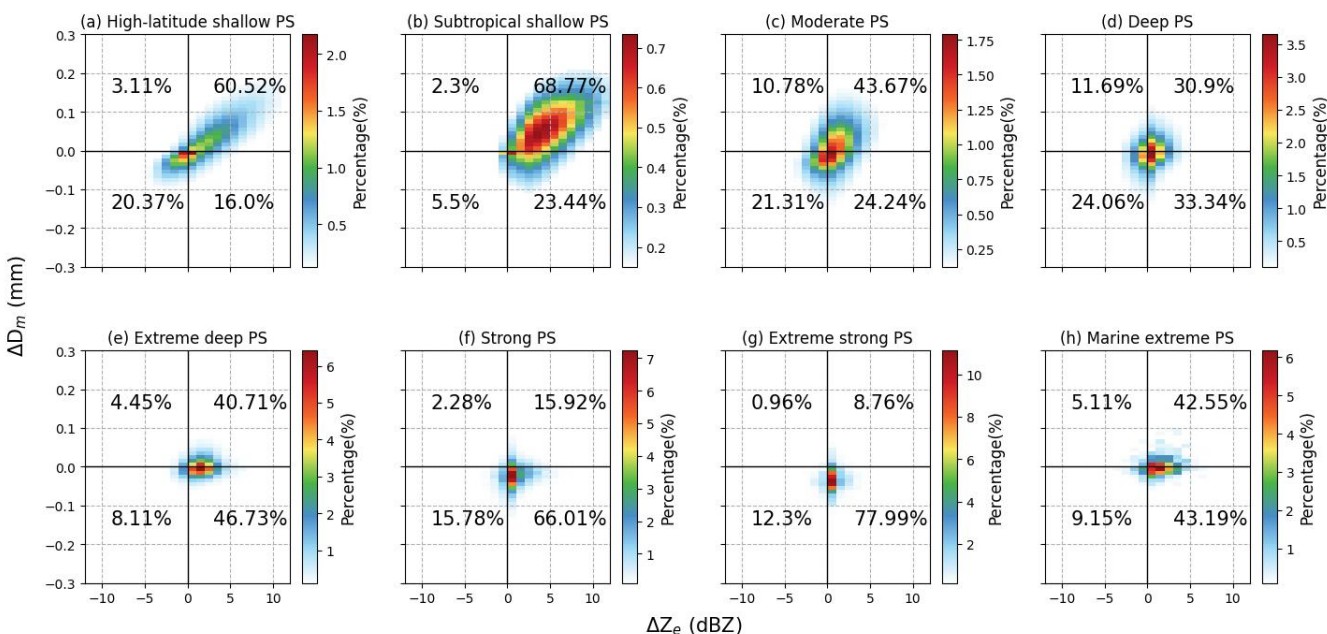


**Figure 8.** Frequency pattern of $\Delta Z_e$ versus $\Delta D_m$ between 2 and 3 km for the eight PS clusters.





### 3.5. Seasonal and diurnal cycles

In this study, seasons were categorized by fixed calendar months. The Northern Hemisphere seasons were defined as spring (March–May), summer (June–August), autumn (September–November), and winter (December–February). Conversely, the Southern Hemisphere seasons followed the opposite pattern: spring (September–November), summer (December–February), autumn (March–May), and winter (June–August). Based on this classification, the subsequent analysis examines seasonal and diurnal variations in PS frequency and microphysical parameters. Figure 9 shows the cycles of PS occurrence. Overall, the seasonal and diurnal cycles differed among the eight types of PSs. Moderate PS, deep PS, strong PS, and extreme strong PS exhibited cycles like those of continental convection systems, with peaks in the afternoon and in summer. Dominated by tropical shallow convection over the ocean (Fig. 1), subtropical shallow PS occurred mostly between 0 and 5 a.m. and was more frequent during the autumn season than during the other seasons, with the lowest occurrence during the spring season. The other types of PS (high-latitude shallow PS, extreme deep PS, and marine extreme PS) did not show obvious diurnal cycles, except that high-latitude shallow PS indicated a low peak at approximately 6 am in winter and a valley before noon in summer. High-latitude shallow PS occurred infrequently in winter. Extreme deep PS occurred more frequently in summer and autumn, with fewer occurrences in winter. Note that marine extreme PS did not demonstrate obvious seasonal discrepancies, but shown a peak at night in the summer. Specifically, strong PS and extreme strong PS with a higher proportion over land exhibit a peak occurrence around 3 p.m. in the afternoon, while extreme deep PS and marine extreme PS with a higher proportion over the ocean shows no distinct peak, with its frequency distributed relatively evenly throughout the day. This difference reflects the land-ocean contrast in extreme PS, which is consistent with findings from other related studies (Wang and Tang, 2020).





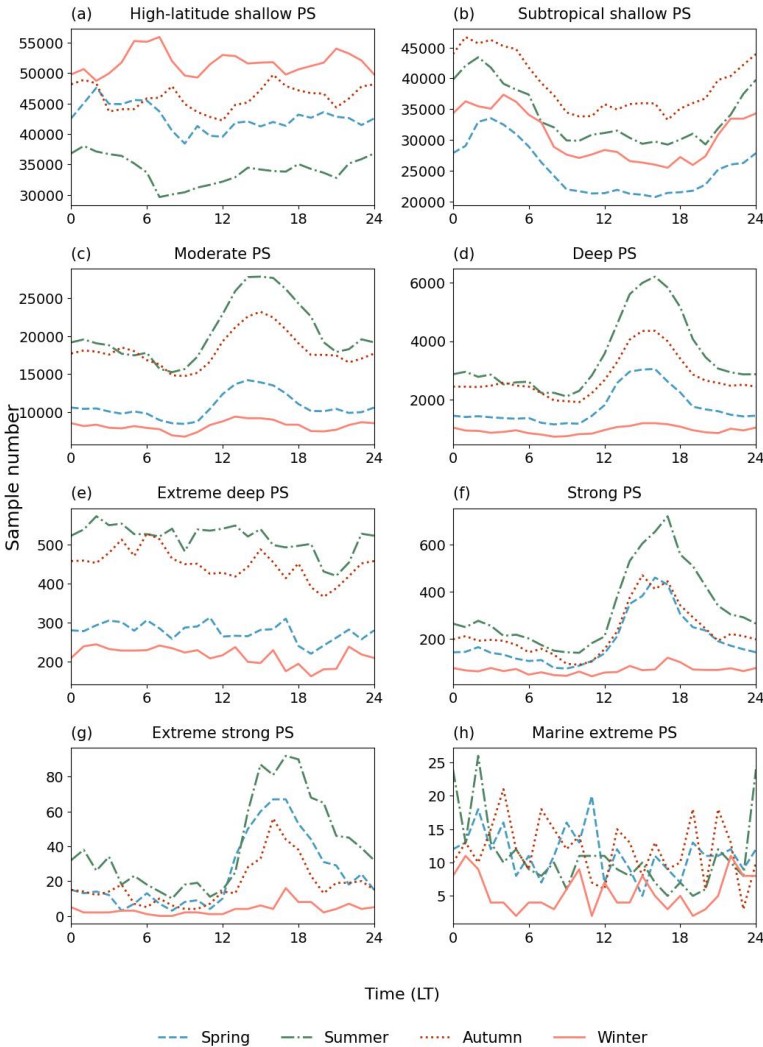

**Figure 9.** Diurnal variations in the sample sizes of the eight distinct PS clusters across the four seasons.

Figures 10 and 11 show the seasonal and diurnal cycles of $D_m$ and $N_w$, respectively. The diurnal cycles of $D_m$ were similar with those of PS occurrence to some extent. For example, in moderate PS, deep PS and strong PS, both the occurrence and $D_m$ have peaks in the around 15 pm. One connection between these two parameters is that environments that favor storm occurrence could also facilitate the development of stronger updrafts, which could promote the formation of large particles in clouds. Nevertheless, discrepancies are obvious between the cycles of occurrence and $D_m$. For example, the $D_m$ in the extreme strong PS did not show obvious diurnal variations. The high-latitude shallow PS shows a peak in the summer (Fig. 10a), which is not found in the diurnal cycle of occurrence (Fig. 9a). In subtropical shallow PS, the diurnal cycle of $D_m$ (Fig. 10b) was the opposite to that of PS occurrence (Fig. 9b). The diurnal cycles of $N_w$ were basically different with those of $D_m$ and occurrence. In subtropical shallow PS, moderate PS, deep PS, and strong PS, the $N_w$ peaked in the morning.





Nevertheless, the diurnal cycles of subtropical shallow PS, moderate PS, and deep PS also differed. For example, $N_w$ of
subtropical shallow PS at night was low, whereas $N_w$ of shallow convective PS and moderate PS at night was very close to
its peak. Extreme deep PS and marine extreme PS did not exhibit obvious diurnal cycles of $N_w$. The extreme strong PS
shown low values of $N_w$ in the afternoon and little variations at night. For high-latitude shallow PS, diurnal variation is not
clear except in the summer when the $N_w$ in the afternoon is the lowest.

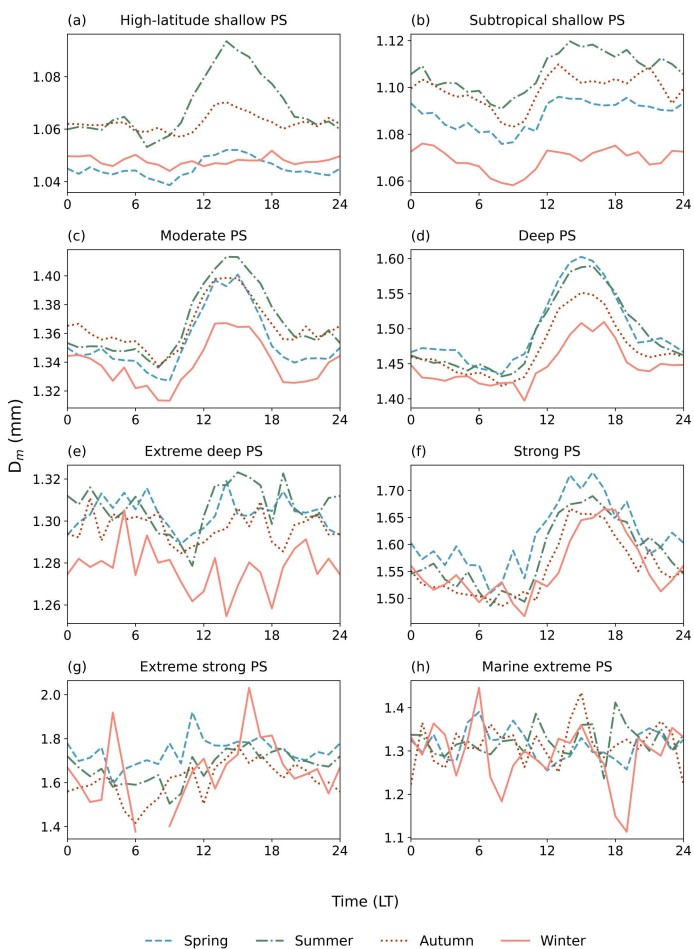


**Figure 10.** Similar to Fig. 9 but for mean $D_m$ value.

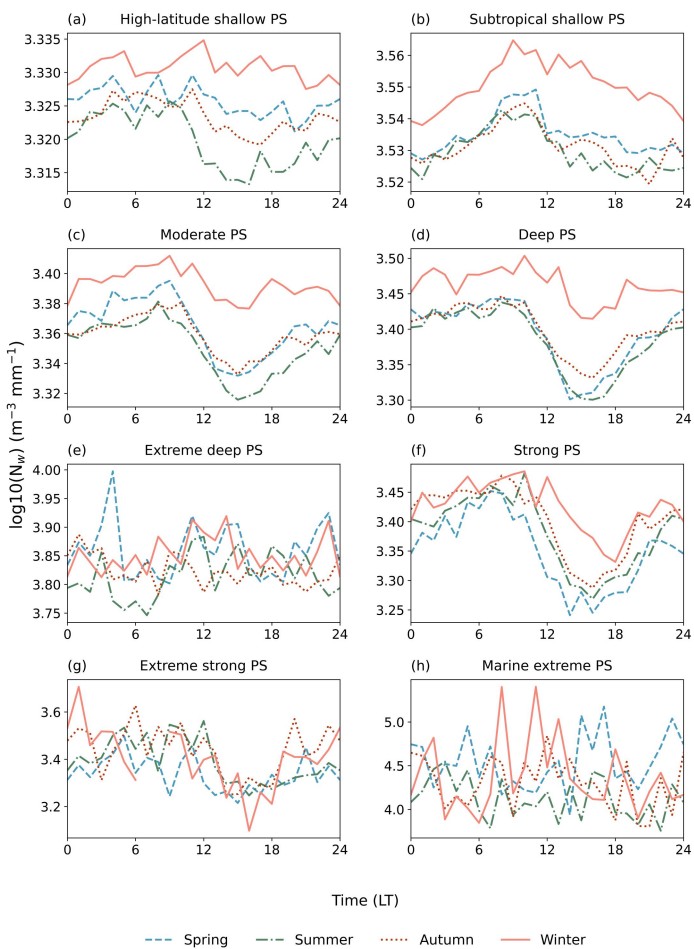

**Figure 11.** Similar to Fig. 9 but for the mean log10($N_w$) value.

Similar to the diurnal cycles, the annual cycles of $D_m$ and $N_w$ were opposite in subtropical shallow PS, moderate PS, and deep PS, of which $D_m$ was the lowest and $N_w$ was the largest in winter. Nevertheless, there were also differences in the annual cycles of the three types of PSs. For example, in subtropical shallow PS, $D_m$ was the largest in summer, followed by autumn and spring, whereas the $N_w$ values during the three seasons were very close. Among the extreme PS, $N_w$ and $D_m$ did not exhibit obvious annual cycles. For high-latitude shallow PS, the highest $D_m$ value occurs in summer and the $D_m$ in winter and spring were comparable. However, the annual cycle of $N_w$ attained the largest value in winter and the lowest value in summer.



## 4. Conclusions

In this study, GPM DPR data were used to objectively classify global PS and analyze the microphysical characteristics of the different types of PS. The main conclusions are as follows:

1). By conducting an objective classification of global PSs via key parameters such as the convective intensity, radar reflectivity, and DSD parameters, eight distinct types of PSs were identified. These systems were classified on the basis of their unique microphysical and convection properties, providing a detailed understanding of the different precipitation processes worldwide. The eight types of PSs identified are as four types of regular/non-extreme PS (high-latitude shallow PS, subtropical shallow PS, moderate PS, deep PS) and four types of extreme PS (extreme deep PS, strong PS, extreme strong PS, marine extreme PS).

2). MAXHT20 is generally correlated with the precipitation rate, but this relationship is not clear for extreme PS. The relationship between MAXHT20 and $D_m$ does not follow a simple linear pattern. For extreme PS, MAXHT20 is positively related to $D_m$ at 2.5 km above the ground surface. This may reflect the relationship between higher cloud tops and greater liquid water contents in strongly convective PSs. However, for non-extreme PS, the relationship between MAXHT20 and $D_m$ is more complex and may be influenced by variations in the physical processes of the different PS.

3). For the same type of PS, $D_m$ over land is greater than that over the ocean. Additionally, $D_m$ exhibits latitudinal variability, particularly in high-latitude shallow PS, where $D_m$ decreases with increasing latitude. Additionally, continental rainfall is associated with lower $N_w$ values due to the cold rain mechanism, whereas oceanic rainfall is associated with higher $N_w$ values resulting from a warm rain regime. Shallow PS generally exhibit narrow distributions of both $D_m$ and $N_w$, particularly in high-latitude shallow PS. Among the strong PS, PS with a higher land proportion exhibit more concentrated $N_w$ values, whereas those with a greater ocean proportion exhibit larger $N_w$ values. However, the distribution of $D_m$ is the opposite: PS with a higher ocean proportion exhibit more concentrated $D_m$ values than land-dominated PSs do.

4). The different PS exhibit distinct microphysical processes. In shallow convective PS, such as subtropical shallow PS and high-latitude shallow PS, coalescence processes largely shape the microphysical characteristics, indicating the aggregation of small raindrops in these PS. In contrast, extreme PSs are characterized by balanced breakup and coalescence processes, highlighting a more complex interaction between raindrop formation and breakup. These results emphasize the varying mechanisms that govern microphysical behavior across the different types of PSs. PS types with high precipitation rates are dominated primarily by balanced breakup and coalescence processes, whereas shallow PSs are characterized mainly by coalescence.

5). The seasonal and diurnal cycles of PSs and their microphysical parameters vary significantly, with distinct patterns observed in different clusters: clusters dominated by continental convection indicate peaks in the afternoon and summer, whereas tropical and high-latitude systems exhibit unique seasonal and diurnal cycles, often with opposite trends between $D_m$ and $N_w$.



Classifying PS is essential for increasing the understanding of the microphysical processes that govern cloud development
and precipitation formation across various climatic regimes. This classification enables the identification of specific
mechanisms that influence rainfall characteristics, such as droplet formation, growth, and distribution, which are vital for
accurate weather predictions and climate modeling. This study revealed the global distribution characteristics of different
types of PS and elucidated the variations in microphysical properties across regions with distinct climatic and geographic
conditions.
In this study, each PS was treated as integrated entity, without considering the variations in $D_m$ and $N_w$ within each system.
Microphysical properties can vary significantly at the pixel level, and relying solely on average or maximum $D_m$ and $N_w$
values captures only part of the overall process. Future work should focus on analyzing pixel-level observations to better
understand the characteristics of microphysical parameters within PS. Furthermore, investigating the relationships between
microphysical parameters and convective parameters will be a key focus of future research. By analyzing the interactions
between these parameters, it is possible to reveal the influences of microphysical characteristics on convective intensity and
precipitation patterns, providing a more detailed perspective for accurately predicting and understanding precipitation
phenomena.
**Data Availability.** The GPM-DPR (version 07A) data from the NASA/G-oddard Space Flight Center are available at
https://disc.gsfc.nasa.gov/datasets/GPM_2A-DPR_07/summary. All statistics and visualization are operated with Anaconda
Individual Edition Python version 3.8.3 (Free Download | Anaconda, accessed on 10 April 2022).
**Author contributions.** XZ and XN conceptualised and planned the research study. XZ conducted the satellite data analysis
with support from XN and drafted the initial manuscript. XN and JZ reviewed and revised the manuscript to refine its
content.
**Competing interests.** The contact author has declared that none of the authors has any competing interests.
**Financial support.** This study is supported by the National Natural Science Foundation of China (42105005), Fundamental
Research Funds for the Central Universities (SWU-KT22007), and General Program of Chongqing Natural Science
Foundation (2022NSCQ-MSX3145).

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
