# Peer review of "Microphysical properties of various precipitation systems worldwide"

_EGUsphere, 2025_

## Author Comment (AC1)

**Reply to community commenter**

Dear community commenter,

We sincerely thank the commenter for the careful reading of our manuscript and for providing detailed and constructive comments. These comments have helped us to re-examine several aspects of the manuscript. In response, we have carefully revised the manuscript to improve its accuracy, clarity, and completeness. Below, we provide detailed responses to each comment raised in the community comment.

**Major comments:**

**Q1:** Why are only the GPM DPR (V07) data from 2018 to 2022 used in the classification study of this paper? Currently, there are more than 10 years of GPM DPR data, and the GPM DPR V07 version algorithm has been applied to the all data before May 2018 (GPM DPR scanning mode changed). The classification algorithm results are closely related to the sample. It is difficult to be convincing if only the GPM DPR data from 2018 to 2022 is used instead of the GPM DPR data of almost all years.

Reply: As the reviewer mentioned, the scanning mode is changed since 2018. This is the key reason why we used the data from 2018 to 2022. During the study period, a total of 8,924,307 precipitation systems occurred, which is a remarkably large number for climatology reanalysis and K-means clustering algorithm. In comparison, the study by Ryu et al. (2021) mentioned by the reviewer only has 328,391 heavy rain events and 6,258,800 heavy rain pixel in their study period 2014 to 2019. In summary, ~ 9 million samples in five years could ensure the statistical robustness of our analysis.

**Q2:** What is the basis for defining the effective precipitation pixel of the precipitation system in this paper as greater than 0.1mm/h? The minimum sensitivity of GPM DPR for detecting precipitation is 0.2mm/h (KaPR) and 0.5mm/h (KuPR). Moreover, in related literature that also uses the definition of precipitation system, greater than 0.5mm/h is used as the standard. This paper uses 0.1mm/h as the selection standard for effective precipitation pixels, which is very likely to introduce unnecessary noise points.

Reply: About the definition of precipitation, we referred to the widely used Precipitation Feature (PF) dataset developed by Liu (2016). The development of PF dataset is also supported by PMM mission and hence we consider it reasonable to carry out our work with reference to this dataset. According to the document of PF dataset, they used near surface precipitation rate > 0 as the threshold for Ku band PF and GMI precipitation rate > 0.1 mm/h for GMI PF. Moreover, similar as the Precipitation Feature dataset, in the process of identifying precipitation system, we only used PS with at least four precipitation pixels, which could significantly reduce noise points.

Meanwhile, many studied using DPR has shown that the minimum rainfall rate of 0.1 mm/h (Peinó et al., 2024; Seela et al., 2024b) and very low mean rainfall intensities (< 0.1

mm/h) are observed (Janapati et al., 2023). Therefore, the threshold 0.1 mm/h is widely used in the applications of DPR. Therefore, we think the threshold 0.1 mm/h is suitable for the study of DPR observation.

**Q3:** The use of the k-means clustering algorithm as a precipitation system classification algorithm does not solve or overcome the inherent defects of k-means, making the results of this study questionable or unreliable. First, combined with Q1, the results presented in this study may change due to changes in the data set. Secondly, in the process of determining the optimal number of categories presented in this study, I questioned: Why can't "11" be the optimal number of samples? In the supporting materials, I found that "11" and "8" both meet the description of the optimal number of samples mentioned in the article. Unfortunately, however, I did not see the reason for excluding "11" in this article.

Reply: As illustrated in the supplementary figure, the DB index attains its lowest value at K=8, whereas the CH score peaks at K=11. This likely accounts for the reviewer's comment that both 8 and 11 may represent optimal cluster numbers. We think that this line of reasoning may not full capture the features of the two indices. When considering the overall trends and variations of DB and CH, it appears more reasonable to select K=8 as the optimal number. Specifically, the DB value reaches its minimum at K=8 and then rapidly rebounds, showing a consistent increase thereafter. In contrast, at K=8 the CH value also attains a relatively high level, after which it remains elevated with fluctuations.

In addition, to further address the reviewer's concern, we also performed clustering with K = 11 and analyzed the corresponding results (Reply-Fig. 1 and Reply-Table 1). A direct comparison shows that the 11-cluster solution largely reproduces the same physical regimes identified in the 8-cluster solution. For example, the new Cluster 2 corresponds closely to the high-latitude shallow PS in the 8-cluster classification, with similar spatial distributions and convective characteristics. Likewise, the new Cluster 10 corresponds to the subtropical shallow PS, the new Cluster 3 to the moderate PS, the new Clusters 4, 7, and 9 to the deep PS, the new Clusters 1 and 5 to the strong PS, the new Cluster 8 to the extreme strong PS, and Cluster 6 to the marine extreme PS.

These comparisons indicate that the 11-cluster solution does not introduce fundamentally new precipitation regimes but rather subdivides existing ones, leading to increased redundancy without providing additional physical insight. Therefore, we adopt K = 8 as the optimal number of clusters, as it captures the major precipitation system types in a more concise and interpretable manner while preserving the essential physical information.

[Figure]

Reply-Figure 1 Spatial distributions (2° × 2° resolution) of the PS counts from 2018 to 2022

Reply-Table 1 Precipitation parameters for the different types of PS

| | Cluster1 | Cluster 2 | Cluster 3 | Cluster 4 | Cluster5 | Cluster 6 | Cluster 7 | Cluster 8 | Cluster 9 | Cluster 10 | Cluster 11 |
|---|---|---|---|---|---|---|---|---|---|---|---|
| Mean MAXHT20 (km) | 14.18 | 2.99 | 5.89 | 10.25 | 15.46 | 12.65 | 12.57 | 16.45 | 8.96 | 2.92 | 13.78 |
| Mean MAXHT30 (km) | 12.87 | 0.66 | 3.86 | 6.97 | 15.29 | 9.15 | 9.71 | 17.01 | 6.36 | 1.32 | 9.52 |
| Mean MAXHT40 (km) | 7.93 | 0.01 | 0.7 | 4.55 | 10.45 | 5.7 | 5.93 | 13.78 | 3.25 | 0.05 | 6.01 |
| Stratiform percentages (%) | 58.06 | 89.14 | 54.53 | 68.49 | 55.84 | 66.89 | 50.47 | 51.44 | 52.37 | 5.57 | 70.81 |
| Convective percentages (%) | 39.29 | 5.67 | 41.29 | 29.28 | 41.40 | 31.54 | 47.24 | 45.60 | 45.34 | 94.28 | 27.44 |
| Land percentages (%) | 60.70% | 20.53% | 23.98% | 19.14% | 73.73% | 10.13% | 49.38% | 84.94% | 34.80% | 6.08% | 12.48% |
| Ocean percentages (%) | 39.30% | 79.47% | 76.02% | 80.86% | 26.27% | 89.87% | 50.62% | 15.06% | 65.20% | 93.92% | 87.52% |
| Mean precipitation (mm h$^{-1}$) | 129.22 | 1.48 | 3.95 | 87.31 | 151.91 | 175.28 | 32.13 | 165.33 | 10.92 | 8.25 | 199.57 |
| Number of samples | 24156 | 3846331 | 1789929 | 60935 | 5176 | 708 | 105906 | 710 | 441863 | 2635780 | 12813 |
| Mean precipitation area (km$^2$) | 9,847,674 | 21,084 | 130,733 | 2,653,996 | 13,098,217 | 7,645,540 | 1,574,530 | 15,019,917 | 404,531 | 39,962 | 8,848,144 |
| >273.15 K frequency (%) | 99.98% | 85.42% | 99.61% | 99.82% | 100.00% | 100.00% | 99.99% | 99.97% | 99.96% | 99.19% | 100.00% |
| 2.5 km Mean MAX-log10($N_w$) [m$^{-3}$ mm$^{-1}$] | 4.67 | 3.43 | 3.9 | 5.09 | 4.83 | 6.13 | 4.25 | 4.89 | 4.16 | 3.68 | 5.3 |
| 2.5 km Mean MAX-$D_m$ [mm] | 3.02 | 0.97 | 1.99 | 2.55 | 3.08 | 2.6 | 2.9 | 3.15 | 2.58 | 1.05 | 2.77 |
| 2.5 km Mean | 3.36 | 3.23 | 3.36 | 3.78 | 3.37 | 4.47 | 3.29 | 3.34 | 3.33 | 3.45 | 3.91 |

| log10(N_w) [m⁻³ mm⁻¹] 2.5 km | | | | | | | | | | |
|---|---|---|---|---|---|---|---|---|---|---|

$\log10(N_w)$ [m$^{-3}$ mm$^{-1}$] 2.5 km

| Mean D$_m$ [mm] | 1.58 | 0.82 | 1.29 | 1.26 | 1.65 | 1.32 | 1.56 | 1.74 | 1.44 | 0.83 | 1.31 |
|---|---|---|---|---|---|---|---|---|---|---|---|

**Q4:** Unfamiliarity with the relevant important literature of this study:

Overall Reply to Q4: We thank the reviewer for the suggestions regarding these papers. We acknowledge that we did not cite all of references mentioned. However, one of them has been cited in our manuscript (Ryu et al., 2021), one was published after our submission (Shi et al., 2025), and one focuses on the impacts of aerosol, which is not directly related to our topic (Xi et al., 2024). Our study aims to reveal the microphysical characteristics of precipitation systems on a global scale, which, to our knowledge, remains insufficiently addressed in the current literature. The majority of previous studies have focused on specific regions. Nevertheless, as the reviewer noted, more and more studies have been emerging recently, such as (Choi et al., 2025; Ryu et al., 2021; Shi et al., 2025). This also indicates that the global microphysics of precipitation is receiving increasing attentions from researchers.

(a) There are important studies that have used GPM DPR data to conduct similar clustering studies; (Such as Global DSD investigation: Ryu, J., Song, H.-J., Sohn, B.-J., & Liu, C. (2021). Global distribution of three types of drop size distribution representing heavy rainfall from GPM/DPR measurements. Geophysical Research Letters, 48, e2020GL090871. https://doi.org/10.1029/2020GL090871 )

Reply: This paper has been cited and discussed in the original manuscript.

(b) Yan Zhang's relevant important papers on the global precipitation system is not mentioned; (Such as: Global precipitation system size, Yan Zhang and Kaicun Wang 2021 Environ. Res. Lett. 16 054005)

Reply: We read this paper during the writing. Nevertheless, we did not cite firstly it because it used the IMERG dataset, but not DPR observation directly. The two kinds of datasets were different. Simply speaking, the IMERG shows the horizontal distribution and the evolution of a storm, while the DPR observation reveal three-dimensional structure and instantaneous observation of a storm. Nevertheless, we cited this pater in the revised manuscript in section 3.1.

(c) One paper, although with a different research purpose, uses both the precipitation system as the basic research object and a similar clustering method. (Aerosol effects on the three-dimensional structure of organized precipitation systems over Beijing-Tianjin-Hebei region in summer)

Reply: We do not aim to relate this paper to aerosol as the relationship between aerosol and precipitation is a complex topic. Meanwhile, our paper aims to reveal the precipitation on a global scale. There are too much regional papers about precipitation microphysics, it is neither practical nor advisable to cite all these references.

(d) This paper does not mention a crucial paper in its investigation of extreme precipitation research. It also uses GPM DPR data and provides valuable conclusions for extreme precipitation research. (A global view on microphysical discriminations between heavier and lighter convective rainfall)

Reply: This paper was published on July, 2025 and we have submitted our manuscript before that. We thank the reviewer's suggestion and have cited this paper in the revised paper.

**Q5:** This study's research method for microphysical processes is relatively simple, considering only the warm rain process, without exploring the contribution of ice phase processes to precipitation and its structure formation.

Reply: To be honest, unlike the numerical model and ground-based observation, previous methods using DPR in the study of microphysical processes is still limited. The method using $\Delta Z_e$ and $\Delta D_m$ was widely used in the literature, such as the study by Shi et al. ( 2025) mentioned by the reviewer in Q4d . Meanwhile, the retrieval process of the dual-frequency radar does not separate the liquid and solid precipitation in its DSD products.

**Q6:** The visualization of this study is relatively simple and not aesthetically pleasing. It does not meet the standards of ACP.

Reply: We have revised the figures according to editors' suggestions in the submission process. If you have any specific suggestions, please let us know and we are pleasure to make revisions. Thanks.

**Detailed comments:**

**Q1:** Some language details are confusing.

(a) In lines 67 and 68, what does PR mean? The PR mentioned above is the abbreviation for precipitation radar. The PR here is obviously not the same, which is confusing.

Reply: We apologize for any confusion. In lines 67 and 68, 'PR' denotes Precipitation Rate, not Precipitation Radar as defined in line 46. This have been revised accordingly in the updated manuscript.

(b) In line 94, what does "if PSs" mean? It is confusing.

Reply: Revised to "of PSs".

(c) There is a duplication of the description of DB index in lines 173 and 175, which is confusing.

Reply: We reworded the context related the introduction of DB index.

**Q2:** There is a lack of common knowledge about GPM DPR radar. The manuscript is very unprofessional in this regard.

(a) The description of the operating band of DPR KuPR in lines 47 and 52 is obviously inconsistent, and the performance parameter values of the operating band should be described to one decimal place.

Reply: The line 47 states that the PR onboard TRMM operated at Ku-band (13.8 GHz). The line 52 indicates that the DPR operate at Ku and Ka band (13.6 GHz and 35.5 GHz). We have revised the values in 52 line from 13 and 35 to 13.6 and 35.5.

(b) The description of lines 53-54 is incomplete. The differential scattering between the two bands caused by rainfall is not only related to the size of the particles, but also to the particle number concentration.

Reply: This sentence was deleted.

(c) In line 54, what do Dm and Nw mean when they appear for the first time? The full text does not provide a detailed definition, and this is the first time its abbreviation appears.

Reply: The full text of $D_m$ and $N_w$ were added here.

(d) Similarly, on line 105, "FS" is not fully described.

Reply: The FS is Full scan. We revised the context.

(e) On line 106, 125 m refers to the vertical range resolution, which is not clearly described here.

Reply: We revised the term to vertical range resolution in the context.

**Reference**

Choi, S., Ryu, J., Lee, S.-M., and Sohn, B.-J.: Characteristics of Global Light Rain System From GPM/DPR Measurements, Journal of Geophysical Research: Atmospheres, 130, e2024JD042434, https://doi.org/10.1029/2024JD042434, 2025.

Janapati, J., Seela, B. K., and Lin, P.-L.: Regional discrepancies in the microphysical attributes of summer season rainfall over Taiwan using GPM DPR, Sci Rep, 13, 12118, https://doi.org/10.1038/s41598-023-38245-z, 2023.

Liu, C.: GPM precipitation feature database (1.0), 2016.

Peinó, E., Bech, J., Polls, F., Udina, M., Petracca, M., Adirosi, E., Gonzalez, S., and Boudevillain, B.: Validation of GPM DPR Rainfall and Drop Size Distributions Using Disdrometer Observations in the Western Mediterranean, Remote Sensing, 16, 2594, https://doi.org/10.3390/rs16142594, 2024.

Ryu, J., Song, H.-J., Sohn, B.-J., and Liu, C.: Global distribution of three types of drop size distribution representing heavy rainfall from GPM/DPR measurements, Geophysical Research Letters, 48, e2020GL090871, https://doi.org/10.1029/2020GL090871, 2021.

Seela, B. K., Janapati, J., Lin, P.-L., Lan, C.-H., and Huang, M.-Q.: Evaluation of GPM DPR Rain Parameters with North Taiwan Disdrometers, Journal of Hydrometeorology, 25, 47–64, https://doi.org/10.1175/JHM-D-23-0027.1, 2024.

Shi, R., Lu, C., Xu, W., and Luo, Y.: A global view on microphysical discriminations between heavier and lighter convective rainfall, Commun Earth Environ, 6, 511, https://doi.org/10.1038/s43247-025-02473-0, 2025.

Xi, J., Li, R., Fan, X., and Wang, Y.: Aerosol effects on the three-dimensional structure of organized precipitation systems over Beijing-Tianjin-Hebei region in summer, Atmospheric Research, 298, 107146, https://doi.org/10.1016/j.atmosres.2023.107146, 2024.

---

## Author Comment (AC2)

**Reply to editor and reviewers**

Dear editor and reviewers,

We sincerely thank you for your valuable and insightful comments, which have greatly helped improve the clarity and quality of our manuscript. In the response, we have carefully revised the relevant sections. We believe these revisions have strengthened the manuscript. Below, we provide detailed responses to each of the reviewers' comments.

**#RC1**

The paper investigates the microphysical properties of global precipitation systems (PSs) using GPM Dual-frequency Precipitation Radar (DPR) data from 2018–2022. An objective k-means clustering approach (with PCA for dimensionality reduction) is applied to classify precipitation systems into eight distinct types:

- Non-extreme PSs: high-latitude shallow, subtropical shallow, moderate, deep.

- Extreme PSs: extreme deep, strong, extreme strong, marine extreme.

  Key findings:

- Continental PSs generally have larger mean drop diameters (Dm) than oceanic PSs, while oceanic PSs have higher normalized intercept parameters (Nw).

- Extreme PSs show balanced raindrop breakup and coalescence, while shallow PSs are dominated by coalescence.

- Clear land–ocean contrasts and latitudinal variations are found in microphysical structures.

- Diurnal and seasonal cycles differ by PS type, with continental systems peaking in the afternoon/summer, while shallow oceanic PSs peak at night.

The manuscript is generally well written and presents novel results. However, it would benefit from a few clarifications and structural improvements. I therefore recommend a minor revision addressing the following points:

Reply: We sincerely thank for the positive evaluation and comments. Accordingly, we have revised the manuscript to improve the logical flow of sections, clarified key methodological details, and ensured that all figures and tables are clearly labeled and referenced.

- The methodology and results section would benefit from a clearer and more logical narrative. At present, the text shifts between clustering, physical interpretation, and microphysical discussion in a way that can confuse the reader. A more transparent structure would be to explicitly present the workflow as follows:

1. Application of clustering algorithm

State clearly that the clustering was applied to the precipitation feature (PF) database, which contains ~9 million PFs identified from GPM DPR data. PF input variables include: precipitation rate, radar echo top heights, drop size distribution (Dm, Nw), convective/stratiform fractions, and spatial metrics.

Reply: We revised the sections 2.3 to include the sample counts and input features. It is worth noting that the parameters are not from the Precipitation Feature (PF) database of Liu et al. (2008), because the PF dataset does not include the profiles of $D_m$ and $N_w$ with 176 levels. Therefore, we reproduced the PS dataset to better derive the required parameters. We referred the new dataset as PS rather than PF to distinguish it from the original PF dataset.

2. Selection of optimal number of clusters

Explain that the Davis–Bouldin index and Calinski–Harabasz score were evaluated for different values of k, and the minimum DB index at k=8 was taken as evidence that eight classes provided the best compromise between compactness and separation. Clarify if the Elbow Method applied to the Within-Cluster Sum of Squares suggests the same number of classes. This will justify the choice of k=8.

Reply: Thank you for this valuable comment. We have also applied the Elbow Method to evaluate the Within-Cluster Sum of Squares (SSE). However, as shown in Reply-Figure 1, the SSE decreases continuously with increasing numbers of clusters, without exhibiting a clear inflection point. Compared with the Davis‐Bouldin index (DBI) and Calinski‐Harabasz score (CHS), the Elbow Method provides less conclusive guidance for determining the optimal number of clusters in our case. Therefore, we relied primarily on the DBI and CHS, both of which consistently indicated that k=8 offers the best compromise between compactness and separation.

[Figure]

Reply-Figure 1. Changes in SSE error, DB index and CH sorce with the number of clusters from 2 to 20.

3.  Characterization of each cluster

Emphasize that each cluster is then characterized by its distinctive features, such as mean Dm or cloud top height... Based on these distinguishing characteristics, the clusters are named descriptively (e.g., "shallow," "deep," "extreme strong," "marine extreme"). This step should be made explicit, because the current version sometimes reads as if the naming were imposed rather than derived.

Reply: We thank the reviewer for this very helpful suggestion. In the revised manuscript, we have moved the description of the naming of the eight clusters to Section 3.1 and clarified the basis for assigning these names, improving the readability and flow of the text.

4.  Emergent spatial and temporal patterns

Only after the clusters are defined should the manuscript show that these objectively derived groups exhibit coherent spatial distributions (e.g., shallow clusters dominating high latitudes, extreme clusters in the tropics). This is an important and interesting result: the clustering, based purely on precipitation properties, also reflects geophysical organisation in space, suggesting that the classification captures physically meaningful regimes.

Reply: Thank you for your valuable comment. We have revised the manuscript and provided a more detailed explanation in Section 3.1. In this section, we clarified both the naming rules of the clusters and their spatial distributions.

- The methodology should also clarify how the stratiform, convective, land, and marine percentages are computed. Are these:
(a) computed for each PF individually and then averaged across all PFs in a cluster, or
(b) computed directly from the total number of pixels across all PFs in a cluster?

Reply: In this study, the percentages of stratiform, convective, land, and marine pixels were first calculated individually for each PS and then averaged across all PSs within a cluster, rather than computed from the total number of pixels across all PSs. We have clarified this point in the corresponding section of the manuscript.

- Extreme events context: The discussion of extreme precipitation systems should be better grounded in previous literature, particularly Zipser et al. (2006), Ni et al. (2017), and Bang and Cecil (2021), which provide benchmarks for extreme convective systems observed by satellites.

Reply: Thank you for this valuable suggestion. We have revised Section 3.1 to strengthen the discussion of extreme convective systems by incorporating relevant findings from previous literature.

- Logical inconsistency (line 287): The argument is circular. Earlier, weak updrafts were inferred from low 40 dBZ echo heights. Later, the rapid decrease in reflectivity with height is attributed again to weak updrafts. This leads to a logical loop and should be clarified.

Reply: Thank you for pointing it. We have clarified the description to avoid circular reasoning. The observations of lower 40 dBZ echo tops and the rapid decrease in reflectivity are now presented as indicators of weaker updrafts, rather than using one to justify the other.

- Balanced processes would result in no change in the reflectivity and Dm.

Reply: Thank you for this helpful comment. We have added this clarification in the revised manuscript, noting that the balanced processes would result in no change in reflectivity and $D_m$.

- Phase-change influence (line 303): The observed change in Dm likely corresponds to changes in precipitation phase across the melting layer. Please reference Mroz et al. (2024). Additional CFADs of Z, Dm, and Nw as a function of temperature or height

relative to the freezing level would strengthen the analysis and could be added as supplementary material.

Reply: Thank you for raising this valuable point. We fully agree that the rapid changes correspond to changes in precipitation phase across the melting layer because of the changes of effective dielectric constant, particle size etc. We added discussion about this point in the section 3.3 and conclusion section. Nevertheless, as we did not include the ERA5 temperature profiles in the Precipitation System dataset, it would take too much time to redownload the DPR dataset and reproduce the dataset. Therefore, we did not plot the CFAD in the temperature coordinate. We will carefully address this issue in future work.

- Algorithm-induced correlations (section 3.4): The observed correlations between Dm and Nw may be artifacts of retrieval assumptions, since the GPM algorithm enforces a correlation between Dm and precipitation rate (see Chase et al., 2020). This must be acknowledged explicitly.

Reply: Thank you for your valuable comment. We have acknowledged in the revised manuscript (Section 3.4) that the observed correlations between $D_m$ and $N_w$ may be partly influenced by algorithm-induced assumptions.

- Extreme precipitation rates: DPR is not well-suited for quantifying extreme rain rates because Ku/Ka frequencies are strongly affected by attenuation and multiple scattering in heavy precipitation. Values above ~100 mm h$^{-1}$ should be treated with caution and interpreted in light of retrieval limitations (see Battaglia et al.).

Reply: We have addressed this point in the Conclusion section of the manuscript. Since the study does not further discuss precipitation rates beyond the quantifications presented in Table 1, we note the limitations of DPR in measuring extreme rain rates, particularly above ~100 mm h$^{-1}$, due to attenuation and multiple scattering.

- Language and Clarity Issues: Many sentences exceed 40 words; shortening them would improve readability. Descriptions of Figs. 7–8 are overly detailed in-text.

Reply: We have revised the manuscript to improve language clarity by shortening long sentences and reducing overly detailed descriptions in Figs. 7–8. Specifically, we have shortened sentences exceeding 40 words and reduced the in-text descriptions of Figs. 7–8.

**RC 2**
**Major comments:**
1. The stated purpose of the analysis is to evaluate distributions of Dm and Nw within precipitation systems that are classified globally using machine learning. A significant flaw in the methodology is that both of these parameters are included as features in the classification algorithms. Thus, the classification of the types is partly (with unknown, possibly varying weight) determined by the parameters that are to be evaluated. The parameters differ amongst the classifications because it was predetermined to be so.

Reply: Thank you for this valuable comment. As mentioned in the last paragraph of the introduction section, this study includes two purposes. The first is to objectively classify

global PSs and the second is to analyze the microphysical characteristics of different PSs. In literature, most papers used subjective methods to recognize various precipitation types, such as use storm height to define deep convection, use precipitation rate to define extreme precipitation etc. Using single threshold is limited because the precipitation systems are quite complex. Therefore, we choose to classify the PS with as much parameters as possible and the k-means clustering was conducted using a combination of multiple precipitation characteristics, including microphysical parameters ($D_m$ and $N_w$), structural parameters ($H_{top}$, MAXHT20/30/40), and macroscopic properties (precipitation area, near-surface precipitation rate, and the convective/stratiform fraction) etc. The subsequent analysis of $D_m$ and $N_w$ is intended to characterize the overall microphysical features of the already-defined PS types, rather than to provide an independent statistical test of $D_m$ – $N_w$ differences among classes.

This approach is popular used in literature. For example, Ryu et al. (2021) used the $D_m$ and $N_w$ to objectively classify three types of heavy rainfall and then discussed the microphysical properties of three types of heavy rainfall. Overall, as $D_m$ and $N_w$ are key descriptors of precipitation microphysics, they are naturally included in both the clustering framework and the physical interpretation of the resulting PS types.

2. To be clear, I am not a radar meteorologist. Generally, though, I believe radar retrievals of microphysical properties, e.g., Dm, would require attention to the fact that the coefficients in the DeltaZe power law relations are not necessarily constant. This would be particularly true in cold clouds where ice particles exhibit various sizes, yes, but also vary in density, habit, and orientation. Cold clouds are definitely sampled here. The methods used to calculate the microphysical parameters presented is largely omitted from this manuscript, as is any justification or explanation of their global applicability. How do you know the differences presented are not artifacts of the retrieval and what are the limitations of the global product?

Reply: We thank the reviewer for the insightful comment. We would like to clarify that the microphysical parameters used in this study, including $D_m$ and $Z_e$, are from the GPM DPR satellite products rather than calculated through our own algorithms. The focus of this study is on applying these existing products to classify and analyze global precipitation systems, rather than developing or validating retrieval algorithms. Besides, amounts of validation work of the GPM DPR products have been widely conducted globally (Adirosi et al., 2021; Chase et al., 2020; Gatlin et al., 2020; Huang et al., 2021; Peinó et al., 2024; Seela et al., 2024). These researches have confirmed that research using these products are reliable.

Similar works using the DPR products are commonly used in the literature; for example, Li et al. (2024) and Wen et al. (2023) also analyzed microphysical parameters from DPR or other satellite products without providing in-depth explanations of the retrieval methodology. In response to the reviewer's concern, we have revised the data description section to explicitly reference the relevant algorithm documentation and literature that describe the retrieval of these parameters in the GPM DPR products.

3. The application of the unsupervised classification is overly complicated. First, there is the PCA. The only reason given to use this is to reduce dimensionality for feature selection supplied to the k-means algorithm. For cloud research, it would be more conventional and straight-forward to reduce the height dimension using an integrating variable, like total water path. There might be some particular reason for using PCs, but none is actually given.

Then, once the clusters are identified, they are (subjectively?) named in such a way that implies they could easily be distinguished with a simple threshold. So why not just apply a geographical and cloud top height threshold to isolate a series of recognizable systems rather than expecting an unsupervised classification to reproduce them?

You might even lose something with the unsupervised approach if cloud types you would want to contrast are actually quite similar. An example of this that appears in the paper are land vs marine systems, which are subset after the fact because the clustering (largely) did not differentiate them.

Reply: We appreciate the thoughtful comments. First, the application of PCA is necessary in this study. The PSs are characterized by multiple variables with vertical structures, and directly including all height-resolved profiles would lead to a very high-dimensional feature space, which would substantially reduce the robustness and efficiency of the k-means algorithm. PCA provides an objective way to retain the dominant modes of vertical variability while filtering out redundant information. Unlike vertically integrated variables (e.g., total water path), which aggregate information and may obscure vertical contrasts, PCA preserves key vertical structure signals that are physically relevant to precipitation processes. Meanwhile, total water path is physically meaningful. In contrast, vertically integrated $Z_e/D_m/N_w$ are rarely used in literature because it is difficult to explain their physical meanings.

Second, the reviewer suggests that simple geographical or cloud-top-height thresholds could be used to isolate recognizable precipitation systems. However, no universally accepted global classification scheme for precipitation systems based on fixed thresholds currently exists. Precious research usually focused on specific precipitation types with one or two precipitation features. However, precipitation characteristics are influenced by multiple interacting factors, including environmental moisture, thermodynamic structure, circulation regime, and topography, all of which vary substantially across regions. A threshold-based approach would therefore introduce strong subjectivity and may fail to capture transitional or mixed regimes that do not conform to predefined criteria.

In contrast, the unsupervised clustering approach integrates multiple precipitation features and objectively obtains various types of precipitation systems. The naming of clusters is performed after clustering, based on their emergent physical characteristics, which does not influence the clustering itself. Meanwhile, the clusters are named with reference to previous literature. While some clusters can indeed be broadly interpreted using simple descriptors (e.g., "shallow" or "deep"), this does not imply that such PSs could be

classified using a single threshold.

We are not sure what the reviewer specifically means by the "*You might even lose something with the unsupervised approach if cloud types you would want to contrast are actually quite similar.*". We would like to clarify that the unsupervised clustering approach in our study does indeed distinguish between continental and marine precipitation systems based on the combined precipitation-related features. As we have known, the discrepancies of precipitation between land and ocean is the one of the most significant characteristics of global precipitation distributions. For example, the hailstorms rarely occur over ocean because the middle level of marine convection tend to be significantly weaker than those over land. The input features of clustering algorithm do not include land-ocean information. However, it separates continental and marine precipitation system well, which proves that the clustering algorithm is reliable.

**Minor comments:**

L54: define terms
Reply: Defined.

L102: So many acronyms. Are all these PR PS MS NS HS FS SSE DB CH CFAD etc really needed? Some like FS are never even defined. Some are defined in the abstract but not elsewhere. Some are defined multiple times and may not be necessary (like PS). Some are defined and hardly used (like CH).
Reply: We appreciate this valuable comment. In the revised manuscript, we have carefully reviewed all acronyms and simplified their usage. For acronyms that are rarely used, we have spelled them out in full spell throughout the manuscript.

L115-116: "those contained the widely" revise.
Reply: Revised.

L135: It's unclear if PSs are individual synoptic systems, cloud systems, or just samples that may have included multiple samples of a given cloud.
Reply: This definition is provided in the first paragraph of Section 2.2. Following the reviewer's comment, we have further clarified the definition of PS in the revised manuscript to avoid confusion regarding the physical meaning of PS.

Sect. 2.3: More details on how PCA was used to augment k-means would be helpful. All parameters or only Ze were put through PCA? How many PCs? How was time-height handled?
Reply: We have revised the section 2.3 to clearly clarify the use of PCA. PCA was applied to the vertical profiles of three parameters: $Z_e$, $D_m$, and $N_w$. The purpose of PCA was to reduce the vertical dimension, compressing the original 176 height levels into a single representative component for each variable. Only one principal component was retained for each profile, as it explains the dominant variance of the

vertical structure and serves as a compact descriptor for clustering. The PCA was applied once, independently for each parameter, and no temporal dimension was involved in the PCA procedure. Time was not treated as an input dimension; only the vertical (height) dimension was reduced.

L140-141: some repetition in these statements
Reply: Revised.

L185-187: Is "extreme" really the right word for the four classes that are described as such? Those clusters are infrequent, but that is not the same as extreme, even if some may have some extreme characteristic amongst or within features. This way of isolating extreme isn't as statistically tractable as analyses of distributions of the cloud water parameters. So it seems like we need to understand extreme in this context means, and why we should be interested in it.
Reply: The term "extreme" is widely used in the atmospheric science community. Generally, an "extreme" event is defined based on its rarity. However, the percentile thresholds used to define extremes vary depending on the context. In the research of extreme precipitation, thresholds such as 1%, 0.1% or even 0.01% percentile are commonly employed. For example, Hamada et al. (2015) selected the uppermost 0.1% of Precipitation Features (PF) to define extreme rainfall events and extreme convective events. In this study, the four types of extreme PS account for 0.39%, 0.22%, 0.023%, and 0.0105% of all PSs. Therefore, it is reasonable to define these low-occurrence PS types as extreme PS.

L212: "where is dominated" a word is missing or something here
Reply: Corrected.

There are lots of unsourced and highly generalized statements (e.g., L253-254, 265-266)
Reply: The statements at L253–254 have been removed in the revised manuscript. The statements at L265–266 are now properly supported with relevant references.

L258: extra space
Reply: Deleted.

**Reference**

Adirosi, E., Montopoli, M., Bracci, A., Porcù, F., Capozzi, V., Annella, C., Budillon, G., Bucchignani, E., Zollo, A. L., Cazzuli, O., Camisani, G., Bechini, R., Cremonini, R., Antonini, A., Ortolani, A., and Baldini, L.: Validation of GPM rainfall and drop size distribution products through disdrometers in Italy, Remote Sensing, 13, 2081, https://doi.org/10.3390/rs13112081, 2021.

Chase, R. J., Nesbitt, S. W., and McFarquhar, G. M.: Evaluation of the microphysical assumptions

within GPM-DPR using ground-based observations of rain and snow, Atmosphere, 11, 619, https://doi.org/10.3390/atmos11060619, 2020.

Gatlin, P. N., Petersen, W. A., Pippitt, J. L., Berendes, T. A., Wolff, D. B., and Tokay, A.: The GPM validation network and evaluation of satellite-based retrievals of the rain drop size distribution, Atmosphere, 11, 1010, https://doi.org/10.3390/atmos11091010, 2020.

Hamada, A., Takayabu, Y. N., Liu, C., and Zipser, E. J.: Weak linkage between the heaviest rainfall and tallest storms, Nat Commun, 6, 6213, https://doi.org/10.1038/ncomms7213, 2015.

Huang, H., Zhao, K., Fu, P., Chen, H., Chen, G., and Zhang, Y.: Validation of precipitation measurements from the Dual-frequency Precipitation Radar Onboard the GPM Core Observatory using a polarimetric radar in South China, IEEE Transactions on Geoscience and Remote Sensing, 60, 1–16, https://doi.org/10.1109/TGRS.2021.3118601, 2021.

Li, D., Qi, Y., and Li, H.: Statistical characteristics of convective and stratiform precipitation during the rainy season over South China based on GPM-DPR observations, Atmospheric Research, 301, 107267, https://doi.org/10.1016/j.atmosres.2024.107267, 2024.

Liu, C., Zipser, E. J., Cecil, D. J., Nesbitt, S. W., and Sherwood, S.: A cloud and precipitation feature database from nine years of TRMM observations, Journal of Applied Meteorology and Climatology, 47, 2712–2728, https://doi.org/10.1175/2008JAMC1890.1, 2008.

Peinó, E., Bech, J., Polls, F., Udina, M., Petracca, M., Adirosi, E., Gonzalez, S., and Boudevillain, B.: Validation of GPM DPR Rainfall and Drop Size Distributions Using Disdrometer Observations in the Western Mediterranean, Remote Sensing, 16, 2594, https://doi.org/10.3390/rs16142594, 2024.

Ryu, J., Song, H.-J., Sohn, B.-J., and Liu, C.: Global distribution of three types of drop size distribution representing heavy rainfall from GPM/DPR measurements, Geophysical Research Letters, 48, e2020GL090871, https://doi.org/10.1029/2020GL090871, 2021.

Seela, B. K., Janapati, J., Lin, P.-L., Chen-Hau, L., and Huang, M.-Q.: Evaluation of GPM DPR Rain Parameters with North Taiwan Disdrometers, Journal of Hydrometeorology, 25, 47–64, https://doi.org/10.1175/JHM-D-23-0027.1, 2024.

Wen, L., Chen, G., Yang, C., Zhang, H., and Fu, Z.: Seasonal variations in precipitation microphysics over East China based on GPM DPR observations, Atmospheric Research, 293, 106933, https://doi.org/10.1016/j.atmosres.2023.106933, 2023.